

# A 1.5-Million-Year Record of Orbital and Millennial Climate Variability in the North Atlantic

David A. Hodell[1], Simon J. Crowhurst[1], Lucas Lourens[2], Vasiliki Margari[3], John Nicolson[1], James E. Rolfe[1], Luke C. Skinner[1], Nicola Thomas[1], Polychronis C. Tzedakis[3], Maryline J. Mleneck-Vautravers[1], Eric W. Wolff[1]

[1]Godwin Laboratory for Palaeoclimate Research, Department of Earth Sciences, University of Cambridge, Cambridge, CB2 3EQ, UK
[2]Department of Earth Sciences, Faculty of Geosciences, Utrecht University, Budapestlaan 4, 3584 CD Utrecht, Netherlands
[3]Environmental Change Research Centre, Department of Geography, University College London, London, WC1E 6BT, UK

*Correspondence to*: David A. Hodell (dah73@cam.ac.uk)

**Abstract.** Climate during the last glacial period was marked by abrupt instability on millennial time scales that included large swings of temperature in and around Greenland (Daansgard-Oeschger events) and smaller, more gradual changes in Antarctica (AIM events). Less is known about the existence and nature of similar variability during older glacial periods, especially during the early Pleistocene when glacial cycles were dominantly occurring at 41-kyr intervals compared to the much longer and deeper glaciations of the more recent period. Here we report a continuous millennially-resolved record of stable isotopes of planktic and benthic foraminifera at IODP Site U1385 (the "Shackleton Site") from the southwestern Iberian margin for the last 1.5 million years, which includes the Middle Pleistocene Transition (MPT). Our results demonstrate that millennial climate variability (MCV) was a persistent feature of glacial climate, both before and after the MPT. Prior to 1.2 Ma in the early Pleistocene, the amplitude of MCV was modulated by the 41-kyr obliquity cycle and increased when axial tilt dropped below 23.5° and benthic $\delta^{18}O$ exceeded ~3.8‰ (corrected to *Uvigerina*), indicating a threshold response to orbital forcing. Afterwards, MCV became focused mainly on the transitions into and out of glacial states (i.e., inceptions and terminations) and during times of intermediate ice volume. During the MPT (1.2-0.65 Ma), obliquity continues to modulate the amplitude of MCV but in a more non-linear fashion as evidenced by the appearance of multiples (82, 123 kyrs) and combination tones (28 kyrs) of the 41-kyr cycle. At the end of the MPT (~0.65 Ma), obliquity modulation of MCV amplitude wanes as quasi-periodic 100-kyr and precession power increase, coinciding with growth of oversized ice sheets on North America and the appearance of Heinrich layers in North Atlantic sediments. Whereas the





planktic δ¹⁸O of Site U1385 shows a strong resemblance to Greenland temperature and
atmospheric methane (i.e., northern hemisphere climate), millennial changes in benthic δ¹⁸O
closely follow the temperature history of Antarctica for the past 800 ka. The phasing of planktic
and benthic δ¹⁸O throughout much of the record is similar to that observed for MIS 3, which
has been suggested to mimic the signature of the bipolar seesaw -- i.e., an interhemispheric
asymmetry between the timing of cooling in Antarctica and warming in Greenland. The Iberian
margin isotopic record suggests bipolar asymmetry was a robust feature of interhemispheric
glacial climate variations for at least the past 1.5 Ma despite changing glacial boundary
conditions. A strong correlation exists between millennial increases in planktic δ¹⁸O (cooling)
and decreases in benthic δ¹³C, indicating millennial variations in North Atlantic surface
temperature are mirrored by changes in deep-water circulation and remineralization of carbon
in the abyssal ocean. We find strong evidence that climate variability on millennial and orbital
scales are coupled across different time scales and interact, in both directions, which may be
important for linking internal climate dynamics and external astronomical forcing.
**1. Introduction**
**1.1 History of Millennial Climate Variability**
Millennial climate variability (MCV) is operationally defined as having a recurrence time
between $10^3$ and $10^4$ years. It excludes variation on orbital timescales but may include
harmonics or combination tones of the orbital cycles that have a period of <10,000 years
(Berger et al., 2006). MCV is part of the background spectrum of climate variability that
follows a power law connecting annual to orbital timescales (Huybers and Curry, 2006). MCV
shows closer relationships to Milankovitch cycles than to higher frequency cycles or
oscillations (Huybers and Curry, 2006) and some MCV may result from non-linear coupling
of processes operating on orbital time scales (Hagelberg et al., 1994). Because climatic
processes are intimately linked across different time scales, documenting the long-term history
of MCV is important for understanding its relationship to orbitally-forced changes in
Quaternary climate.
The first millennial event to be widely recognized in paleoclimate records was the Younger-
Dryas when a 1,300-yr-long period of cold climate began at 12,800 yrs BP and reversed the
general warming trend of the last deglaciation in the Northern Hemisphere (for a review, see
Mangerud, 2021). Further study of Greenland ice cores revealed the common occurrence of
similar abrupt warming/cooling events during Marine Isotope Stage (MIS) 3 (~57 to 29 ka).



These Dansgaard-Oeschger (D-O) events represent the rapid switching of North Atlantic
climate between colder stadial and warmer interstadial states in less than 100 years with a
recurrence time of ~1500 years (Dansgaard et al., 1982). The discovery of such abrupt climate
changes in Greenland in the early 1980s was unexpected because of the great magnitude and
rapidity of the temperature change and short recurrence times.

Following the recognition of MCV in Greenland, the search began to see if similar events were
recorded in marine sediment cores in the North Atlantic. Marine evidence for D-O events was
found in variations in sediment color and the abundance of the polar foraminifer
*Neogloboquadrina pachyderma* (sinistral) at DSDP Site 609 (Broecker et al., 1990; Bond et
al., 1992, 1993). During some of the most extreme stadial events, North Atlantic marine
sediment cores were also found to contain layers of ice-rafted detritus (IRD) that are rich in
detrital carbonate derived from Paleozoic bedrock underlying Hudson Strait (Heinrich, 1988;
Broecker et al., 1992; Hemming, 2004). These so-called 'Heinrich events' were attributed to
massive discharges of the Laurentide Ice Sheet to the North Atlantic via Hudson Strait. The D-
O cycles are packaged into longer-term cycles ("Bond cycles") where the amplitude and
duration of stadial-interstadial events decrease as climate become progressively cooler until
terminating in a Heinrich stadial, which is followed by a large abrupt warming (Bond et al.,
1993). The recurrence time of Bond cycles and Heinrich events is on the order of every ~7-8
kyrs, which is longer than D-O events.

MCV, as expressed in Greenland temperature, has a counterpart variation in Antarctic ice cores
that is smaller in magnitude and more gradual in nature than the signals found in Greenland.
The one-to-one coupling between these events is often explained by changes in inter-
hemispheric heat transport referred to as the thermal bipolar seesaw (Bender et al., 1994;
Stocker, 1998; Blunier and Brook, 2001; EPICA Community Members, 2006; WAIS Divide
Project Members, 2015). The duration of stadials in Greenland is linearly correlated with the
strength of warmings in Antarctica (EPICA Community Members, 2006; WAIS Divide Project
Members, 2015). The longer-duration interstadials in Antarctica (Antarctic Isotope Minimum
or AIM events) are also marked by rises in atmospheric $CO_2$ (Ahn and Brook, 2014; Bauska et
al., 2021), presumably from decreased stratification and increased overturning in the Southern
Ocean (Anderson et al. 2009; Skinner et al., 2010, 2020). On millennial time scales, $CO_2$
closely tracks Antarctic temperature with peak $CO_2$ levels lagging peak Antarctic temperature
by more than 500 years (Bauska et al., 2021). The magnitude of the $CO_2$ rise is correlated with

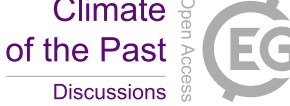

the duration of the North Atlantic stadial stage (Buizert and Schmittner, 2015), with a greater
$CO_2$ response during times of prolonged stadial conditions in Greenland, such as those
associated with Heinrich events. These longer-lived millennial events represent major
reorganizations of the ocean-atmosphere system and have far-reaching effects well beyond the
North Atlantic region.

A leading hypothesis is that changes in deep-water/ocean circulation have played a key role in
MCV (for review, see McManus et al., 2004; Alley et al., 2007; Henry et al., 2016; Lynch-
Stieglitz, 2017; Menviel et al., 2020). The Atlantic Meridional Overturning Circulation
(AMOC) is sensitive to mode jumps that can be triggered by changes to the surface-water
density in North Atlantic source areas of deep-water formation. Climate models of varying
complexity have simulated millennial oscillations when forced by freshwater fluxes from
melting ice (Stocker and Johnsen, 2003; Ganopolski and Rahmstorf, 2001; Timmermann et al.,
2003; Rahmstorf et al., 2005), whereas others have emphasized the role of sea ice (Gildor and
Tziperman, 2001; Sevellec and Fedorov, 2015; Li et al., 2005, 2010) and/or ice shelf dynamics
(Dokken et al., 2013; Petersen et al., 2013). Some model simulations have shown spontaneous
oscillation of the AMOC even in the absence of deliberate fresh-water forcing (Winton and
Sarachik, 1993; Sakai and Peltier, 1999; de Verdière, 2007; Kleppin et al., 2015). Others have
implicated orbitally-induced insolation changes or variations in atmospheric $CO_2$ as (external
to the North Atlantic) triggers of MCV (Friedrich et al., 2010; Zhang et al., 2021; Yin et al,
2021; Zhang et al., 2017; Vettoretti et al., 2022).

Oxygen isotope records of foraminifera capable of resolving orbital-scale variations are
numerous (for a summary of records and resolutions, see fig. 2 of Ahn et al., 2017), but few
long millennial-resolved records exist to examine the interaction between orbital and millennial
components of the climate system. The study of long-term changes in MCV requires long
continuous sedimentary sequences with high sedimentation rates from climatically sensitive
areas of the world ocean. Some marine records of MCV exist beyond the last glacial cycle
(McManus et al., 1999; Hodell et al., 2008; Oppo et al., 1998; Kawamura et al., 2017; Jouzel
et al., 2007; Loulergue et al., 2008; Barker et al., 2011, 2015; Martrat et al., 2007; Margari et
al., 2010; Alonso-Garcia et al., 2011; Burns et al., 2019; Gottschalk et al., 2020), but only a
few extend beyond 800 ka into the early Pleistocene (Raymo et al., 1998; McIntyre et al., 2001;
Birner et al., 2016; Billups and Scheinwald, 2014; Hodell et al., 2008; Hodell et al., 2015;
Hodell and Channell, 2016; Barker et al., 2021, 2022).




Here we present a 1.5-million-year record of millennial variability in surface- and deep-water
properties as recorded by stable isotopes of planktic and benthic foraminifera at IODP Site
U1385 (the "Shackleton Site") located off Portugal in the NE Atlantic Ocean (Fig. 1). The
Iberian margin is a well-known location for sediment cores that capture orbital- and millennial-
scale variations in North Atlantic climate (Shackleton et al., 2000; 2004; Martrat et al., 2007;
Hodell et al., 2013, 2015). Because of its location in the eastern Atlantic at ~37°N, the region
is sensitive to migrations in the Polar Front but is positioned far enough south that proxies don't
saturate under full glacial or interglacial conditions.

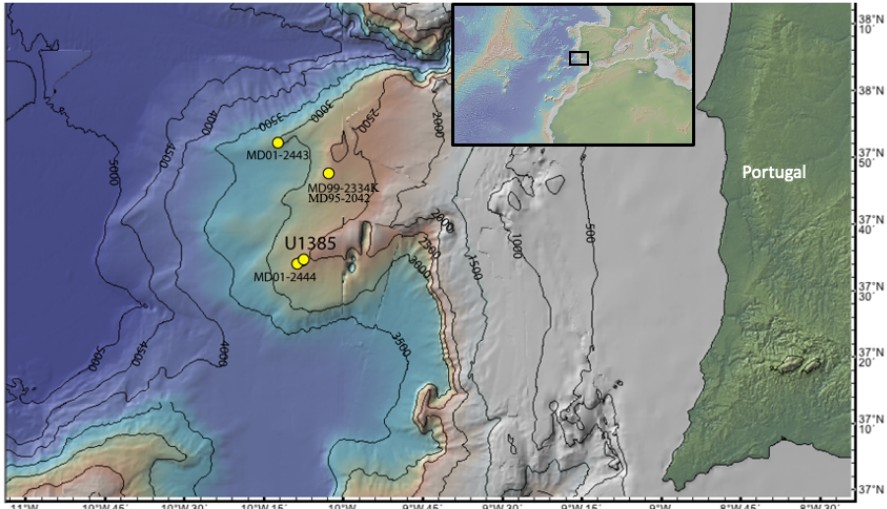

**Figure 1**. Location of IODP Site U1385 and selected piston (MD95-2042, MD01-2444,
MD01-2443) and kasten (MD99-2334K) cores on the Promonotorio dos Principes de Avis,
along the continental slope of the southwestern Iberian margin. The map was made with
GeoMapApp (www.geomapapp.org) using bathymetry of Zitellini et al. (2009).

The long, millennial-resolved isotope records from Site U1385 provide an opportunity to
address several questions about the nature of MCV on orbital and millennial timescales. How
common was MCV during older glacial periods of the Pleistocene? Does the nature (intensity,
duration, pacing) of MCV change with orbital configuration or climate background state (ice
volume, sea-level, ice sheet height)? What is the relationship between MCV and longer-term,
orbitally-driven glacial-interglacial cycles – how do they interact? How did MCV change
across the Middle Pleistocene Transition (MPT) when ice sheets grew larger in size and the
amplitude of glacial-interglacial cycles increased? Was the thermal bipolar seesaw mechanism



active during older glacial periods of the Pleistocene? What role did millennial variability play
in atmospheric $CO_2$ variations or vice-versa?

**1.2 The Iberian Margin Record**

The Greenland and Antarctic ice cores provide continuous paleoclimate records to ~123,000
(NGRIP Project Members, 2004) and 800,000 years (Jouzel et al., 2007) before present,
respectively. Beyond the age of the oldest ice, we must rely upon rapidly accumulating marine
sediments to document the older history of short-term climate variability in the North Atlantic.
Piston cores from the Iberian margin off Portugal contain clear signals of D-O variability in
marine sediments (Shackleton et al., 2000, 2004; Martrat et al., 2007; Margari et al., 2010,
2020). High accumulation rates provide the temporal resolution needed to capture the relatively
brief, abrupt temperature changes observed in the Greenland ice core. Shackleton et al. (2000,
2004) demonstrated that each of the D-O events in Greenland is expressed in the Iberian margin
planktic $\delta^{18}O$ signal over the last glacial cycle (Fig. 2). In the same sediment core, the benthic
$\delta^{18}O$ signal resembles the $\delta D$ record in Antarctic ice cores (Shackleton et al., 2000, 2004),
capturing each of the Antarctic Isotope Maximum (AIM) events (Jouzel et al., 2007). Because
the influence of both Greenland and Antarctic millennial events is co-registered in the same
sediment core, the phasing can be determined stratigraphically without the usual limitations
associated with determining the absolute ages of short-lived climate events. The observed
phasing of isotope signals for the last glacial cycle is consistent with the relative changes in
temperature between Antarctic and Greenland deduced from the synchronization of ice core
records using methane (Fig. 2) (Blunier and Brook, 2001; WAIS Divide Project Members,
2015). This pattern has been interpreted as a manifestation of the thermal bipolar seesaw
(Stocker and Johnsen, 2003) and can be used to recognize a similar mode of operation of the
ocean-climate system in older ice cores (Loulergue et al., 2008) and Iberian margin sediment
cores (Margari et al., 2010).

The benthic $\delta^{13}C$ signal of deep cores from the Iberian margin provides a record of changes in
the $\delta^{13}C$ of deep-water dissolved inorganic carbon (DIC), which varies with changes in deep-
water source areas, mixing of water masses, and oxidation of organic matter once the water
mass is isolated from the surface ocean. In Iberian margin piston cores, surface cooling is
associated with systematic decreases in benthic carbon isotopes, indicating concomitant
changes in North Atlantic surface temperature and deep-water circulation (Martrat et al., 2007).
Cooling is associated with a shoaling of the Atlantic overturning cell that results in a decreased



influence of high-$\delta^{13}$C North Atlantic Deep Water (NADW) and an increase of southern-
sourced waters with low $\delta^{13}$C at abyssal depths in the North Atlantic.

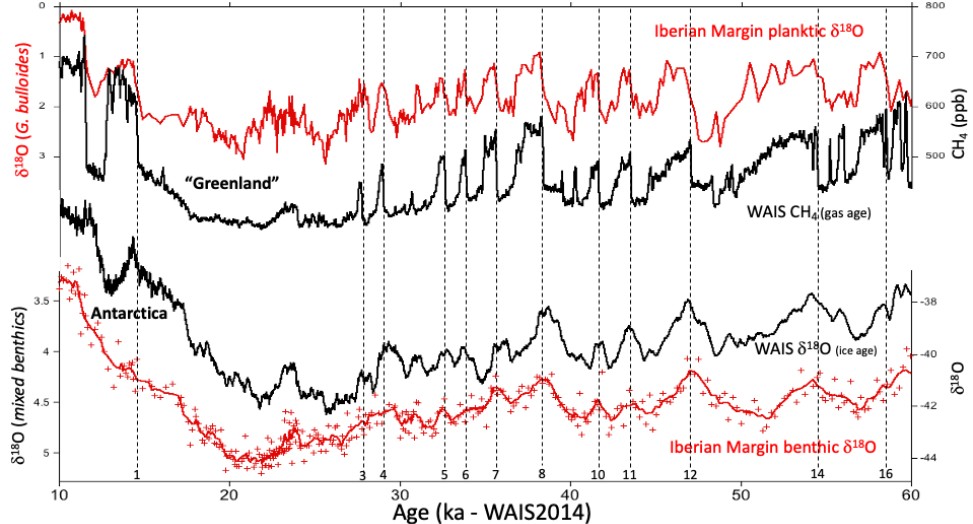


**Figure 2**. Comparison of Iberian margin $\delta^{18}$O records and polar ice cores. Top panel: Planktic
$\delta^{18}$O from core MD95-2042 (Shackleton et al., 2000) compared with CH$_4$ from the WAIS
Divide ice core on Antarctica (WAIS Divide Project Members, 2015); Bottom panel: benthic
$\delta^{18}$O in core MD95-2042 compared with the $\delta^{18}$O record of the WAIS Divide ice core (WAIS
Divide Project Members, 2015). Vertical dashed lines are drawn at the abrupt transitions from
cold stadials to warmer interstadial conditions in Greenland and are numbered at the bottom of
the figure. Note that the phasing of planktic and benthic $\delta^{18}$O is the same as that inferred from
the CH$_4$ and $\delta^{18}$O in the WAIS Divide ice core. This pattern has been interpreted as being
indicative of a thermal bipolar seesaw.

Because of the relative sensitivity of surface and deep-water signals on the Iberian margin to
millennial climate change, this area was targeted by the International Ocean Discovery
Program (IODP) to extend the record beyond the oldest piston cores from the region. In 2011,
five holes were drilled at IODP Site U1385 (the "Shackleton site") off Portugal, resulting in
the recovery of a continuous 166.5-m sequence. A composite section was constructed by
correlating elemental data measured by core scanning XRF at 1-cm resolution in all holes
(Hodell et al., 2015). The U1385 record extends to 1.45 Ma (MIS 47) with an average
sedimentation rate of 11 cm kyr$^{-1}$ (Hodell et al., 2013; 2015). The record is mostly complete





except for a short hiatus at Termination V that has removed part of late MIS 12 and early MIS
11 (Oliviera et al., 2016).

**2. Materials and Methods**

**2.1 IODP Site U1385 ("Shackleton site")**

Site U1385 is located very near the position of piston core MD01-2444 (37° 33.88′ N, 10° 8.34′
W, 2656 m below sea level; Fig. 1), which consists of a 27-m long sequence representing the
last 194 kyr of sediment deposition. Core MD01-2444 can be precisely correlated to Site
U1385 on the basis of Ca/Ti measured every 1-cm in both cores (Hodell et al., 2015), thereby
providing an equivalent depth (crmcd) in Site U1385 corresponding to each depth in core
MD01-2444. Placing MD01-2444 on the Site U1385 depth scale corrects for the well-known
effects of streching and compression that may affect cores recovered with the jumbo Calypso
coring system (Skinner and McCave, 2003).  Because we did not measure stable isotopes for
the upper 23 m of Site U1385 at high resolution, the isotope records presented here consist of
a splice between core MD01-2444 (Vautravers & Shackleton 2006; Margari et al., 2010;
Hodell et al., 2013; Tzedakis et al., 2018) and Site U1385 (this study). The U1385 record is
appended to MD01-2444 at 27.45 m in the piston core which is equivalent to 26 crmcd in Site
U1385, corresponding to an age of ~194 kyrs.

Oxygen and carbon isotope measurements of planktic and benthic foraminifera from Site
U1385 were made at an average temporal resolution of ~200 years for the last 1.45 million
years (Fig. 3). The analytical methods were similar to those described by Hodell et al. (2015).
For planktic foraminifera, we used the surface-dwelling species *Globigerina bulloides* from
the 250 - 350 um size fraction. We interpret the millennial variations in plankic $\delta^{18}$O of *G.*
*bulloides* as reflecting variations in sea surface temperature (SST) in the NE Atlantic, which is
supported by the strong inverse correlation of planktic $\delta^{18}$O and alkenone SST data from
Iberian margin cores for the past 400 ka (Martrat et al., 2007). For benthic foraminifera, we
used mostly *Cibicidoides wuellerstorfi* and occasionally other species of *Cibicidoides* from the
>212 um size fraction. In samples where specimens of *Cibicidoides* spp. were absent, we used
$\delta^{18}$O of *Uvigerina peregrina* or *Globobulimina affinis*. All $\delta^{18}$O values for each species were
corrected to *Uvigerina* using the offsets suggested by Shackleton et al. (2000) -- i.e., +0.64 for
*Cibicidoides* and -0.3 for *G. affinis*. We recognise these offset may vary slightly with time
(Hoogakker et al, 2010) but are not large enough to affect the pattern of benthic $\delta^{18}$O variation.






**Figure 3.** δ18O (per mil, VPDB) of the planktic foraminifer *Globigerina bulloides* (top) and benthic foraminifer *Cibicidoides wuellerstorfi* (bottom) at IODP Site U1385. Interglacial Marine Isotope Stages (MIS) are labeled with odd numbers in top panel and glacial stages with even numbers in bottom panel. Photo credit: *G. bulloides* (https://www.mikrotax.org/pforams/index.php?id=104034), *C. wuellerstorfi* (http://foraminifera.eu/single.php?no=1000394&aktion=suche)



The water depth of Site U1385 (2578 meters below sea level) places it under the influence of
Northeast Atlantic Deep Water today but it was influenced by southern sourced waters during
glacial periods. Variations in benthic $\delta^{18}$O reflect changes in temperature and the $\delta^{18}$O of
deep water bathing the site, which was affected by ice volume on orbital time scales, albeit
with such ice-volume signals being transported to the core sites on the timescale of ocean
mixing (Duplessy et al., 1991; Skinner et al., 2005; Waelbroeck et al., 2011). Millennial
variations in benthic $\delta^{18}$O are affected by changes in deep-water temperature and by the
watermass endmember isotopic compositions (Shackleton et al., 2000; Skinner and
Elderfield, 2003; Skinner et al., 2003, 2007). For benthic $\delta^{13}$C, we use only the data from the
epibenthic *C. wuellerstorfi* to monitor changes in deep-water ventilation related to changes in
deep ocean circulation and remineralization of organic carbon.
Core scanning XRF measurements were made every 1 cm in piston core MD01-2444 (Hodell
et al., 2013) and all holes drilled at Site U1385 (Hodell et al., 2015). The Ca/Ti signal was used
to correlate among holes and define a composite spliced section consisting of intervals from
Holes A, B, D and E to form a total length of 166.5 m. The spliced section used in this study
consists mostly of Holes D and E with a few sections taken from Holes A and B to bridge core
gaps. All sample depths are given in corrected revised meter composite depth (crmcd) that are
corrected for stretching and squeezing caused by coring distortion (Pälike et al., 2005).
Theoretically, the same crmcd should be equivalent in all holes but, in practice, the accuracy
of the alignment among holes is dependent upon the scale of the correlative features and
variability of the Ca/Ti record. We estimate that Ca/Ti features are correlated to the decimeter
level or better.
Orbital and millennial variability at Site U1385 is expressed in sediment compositional changes
as reflected by elemental ratios (Hodell et al., 2013, 2015). Detrital sediment supply increases
relative to biogenic production during cold periods, which is reflected in an increase in Zr/Sr
and decrease in Ca/Ti (Hodell et al., 2015), which are inversely correlated with one another.
During the last glacial cycle, increases in Ca/Ti occur during Greenland interstadials whereas
peaks in Zr/Sr mark the stadials, particularly those containing Heinrich events (Channell et al.,
287   2018).





**2.2 Chronology**

We have updated previous age models of piston core MD01-2444 and IODP Site U1385
(Hodell et al., 2013, 2015) and provide several alternative time scales so users can choose the
chronology that is best suited to their specific application. The age models for MD01-2444
include (0 to 194 ka): (1) WAIS Divide (WDC2014) by correlation of planktic $\delta^{18}O$ to WAIS
methane between 10 and 60 kyrs; (2) AICC 2012 for MD01-2444 by correlation of benthic
$\delta^{18}O$ to $\delta D$ of EPICA from 60 to 135 ka and using the tie points of Shin et al. (2020) from 135
to 190 ka during MIS 6; (3) a Corchia speleothem chronology is provided for MIS 5 by
correlation of planktic $\delta^{18}O$ to the $\delta^{18}O$ of the stalagmite record (Tzedakis et al., 2018).

The age models from MD01-2444 (0 to 194 ka) are combined with those for Site U1385 (>194
ka) to produce the following chronologies: (1) AICC2012 to 800 ka by iteratively correlating
millennial events in Site U1385 planktic $\delta^{18}O$ to EPICA $CH_4$ (gas age) and benthic $\delta^{18}O$ to
EPICA $\delta D$ (ice age), (2) Greenland Synthetic (0-800 ka) by correlation of the planktic $\delta^{18}O$ to
Barker et al. (2011), (3) revised LR04 chronology (Lisiecki and Raymo, 2005) based on
correlation of Site U1385 benthic $\delta^{18}O$ to the Prob Stack (0 to 1450 ka) (Ahn et al., 2007), and
(4) an orbitally-tuned time scale by correlation of L* to the Mediterranean sapropel stratigraphy
of the eastern Mediterranean (Konijnendijk et al., 2015). In general, the tuned time scale of
Site U1385 compares favorably with LR04 within the estimated error of the chronology, which
is ±4 kyr for the past million years and ±6 kyr for the interval from 1.0 to 1.5 Ma (Lisiecki and
Raymo, 2005).

The chronology used in this paper is a hybrid model constructed using a combination of age-
depth points from MD01-2444 and U1385. The age model is accurate to a precession cycle
(~23 kyrs) but cannot provide exact absolute or relative dates for millennial events. This
shortcoming limits the reliability of suborbital spectral peaks and estimation of recurrence
times of millennial events. Nonetheless, the relative phasing of signals recording different
components of the ocean-atmosphere system can be determined stratigraphically without the
need for a time scale that is accurate at suborbital resolution. This is particularly important for
inferring the phase relationship between planktic and benthic $\delta^{18}O$, which reflects the
interhemispheric leads and lags of the two polar regions.



## 3. Results

### 3.1 Defining millennial variability

To identify millennial events, it is necessary to isolate the high-frequency component of the record by eliminating the low-frequency variations related to direct orbital forcing. We experimented with several methods for accomplishing this task including high-pass filtering, Gaussian smoothing of the record followed by calculation of a residual, and subtracting the planktic and benthic $\delta^{18}O$ values from one another. Although there are subtle differences in detection of millennial events depending on the method and thresholds used, the fundamental identification of millennial events was similar among methods. For simplicity, we settled on a high-pass Butterworth filter of second order with a cutoff frequency starting at 1/20 ky. The data were interpolated to equal time steps of 0.2 ka prior to filtering.

We identified stadial and interstadial events using the 'findpeaks' function in MatLab by specifying a peak height that must exceed a threshold defined by a multiplier of the standard deviation of the data (e.g., $1\sigma$ or $1.5\sigma$), and a minimum peak duration and recurrence time (1 kyr). We varied the parameters so that the algorithm correctly identifies all known D-O events for the last glacial cycle in Core MD01-2444. The same parameters are then applied to identify millennial events for the entire length of the record.

There is some degree of subjectivity involved in identifying millennial events. If the same event is identified in both the planktic $\delta^{18}O$ and Zr/Sr signals (Figs. 4 and 5), we can be confident the event is robust; however, this is not always the case. Not every millennial event in planktic $\delta^{18}O$ has a corresponding change in Zr/Sr, which preferentially records the strongest of the stadial events. Additionally, the planktic $\delta^{18}O$ record can miss stadial events associated with glacial terminations (i.e., terminal stadial event) because the decrease in the $\delta^{18}O$ of seawater from melting ice overwhelms the $\delta^{18}O$ increase expected from cooling. In this case, we rely on the increase in Zr/Sr to recognize the event. Most terminal stadial events are also associated with a minimum in benthic $\delta^{13}C$ that can be used as an ancillary indicator of these events near glacial terminations. Forthcoming high-resolution measurements of the alkenone SST proxy at Site U1385 will greatly improve the identification of millennial events, especially those associated with terminations (Rodrigues et al., 2017).





We summed the number of millennial events (stadials + interstadials) over a moving non-
overlapping window of 10-kyr for both planktic δ¹⁸O and Zr/Sr. Patterns of millennial
variability were similar for the two proxies (Figs 4 and 5). The number of events per 10-kyr
interval changes depending upon the choice of start time of the 10-kyr window and whether
the analysis is run forward or backwards, but the fundamental patterns are not substantially
altered. The greatest number of millennial events per 10-kyr interval occurred during MIS 3
and glacial stages of the early Pleistocene from MIS 38 to 46.

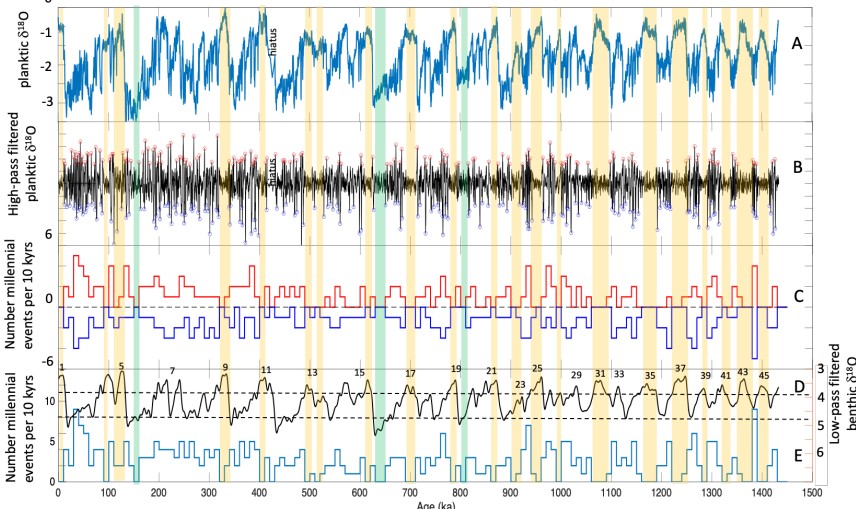

**Figure 4**. (A) The δ¹⁸O record of *G. bulloides* at Site U1385. (B) High-pass filter of (A) to
remove orbital frequencies and extract suborbital variability. Stadial (blue circles) and
interstadial (red circles) events are identified by values that are greater than 1 standard
deviation from the mean. (C) The number of stadial (blue) and interstadial (red) events in non-
overlapping windows of 10,000-year duration. (D) Low-pass filter of benthic δ¹⁸O record
(black) used to lookup δ¹⁸O values for each millennial event. Horizontal dashed black lines
correspond to the benthic δ¹⁸O thresholds marking the window of enhanced millennial
variability. (E) The number of millennial events is the sum of the stadial and interstadial events
in (C). The orange shade indicates times when there are no millennial events per 10,000 years
associated with full interglacial stages. Green shade indicates where there are no millennial
events per 10,000 years associated with full glacial stages.



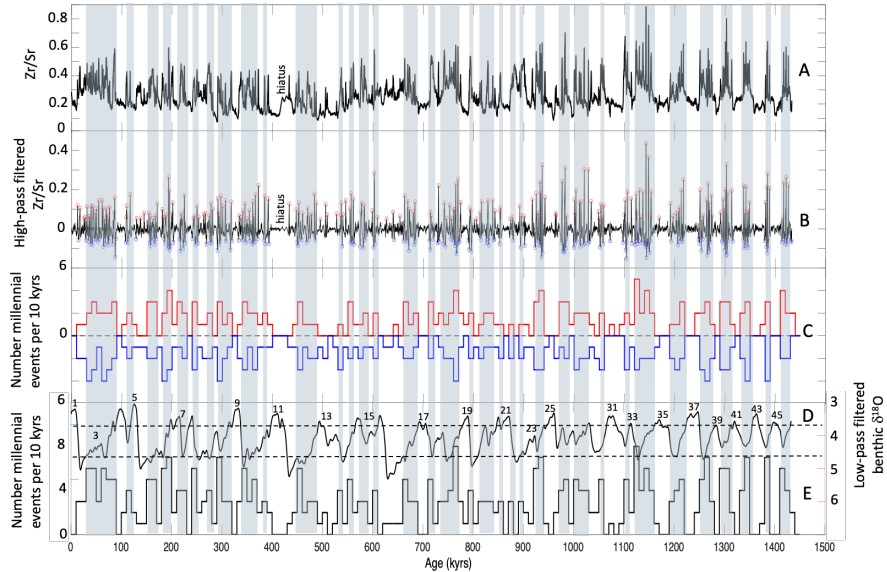

**Figure 5**. Same as Figure 4 but for Zr/Sr. The blue shade indicates times when the total number of millennial events equals or exceeds 3 per 10,000 years, which occurs mostly during intermediate glacial states.

## 3.2 Power Spectra

Time series analysis was performed using Acycle (Li et al., 2019) in the MatLab environment. The power spectra of planktic $\delta^{18}O$ and Zr/Sr show significant peaks at ~100 ka, 41 ka and 23 ka in the orbital band (Fig. 6). The suborbital part of the spectrum is complex with many high-frequency peaks. The spread of frequencies may partly reflect error associated with the chronology that smears the concentrations of any spectral energy across a wide band of frequencies (Rhines and Huybers, 2011). We have more confidence in spectral peaks that occur in both isotopic and lithologic records but spectral estimates are highly susceptible to sedimentation rate changes and age model errors. We reognize a band of peaks between 4 and 6 kyrs and many peaks with periods less than 3 kyrs. This is generally consistent with observations of a grouping of millennial events for MIS 3 into those with longer (Bond cycles) and shorter (D-O events) recurrence times.

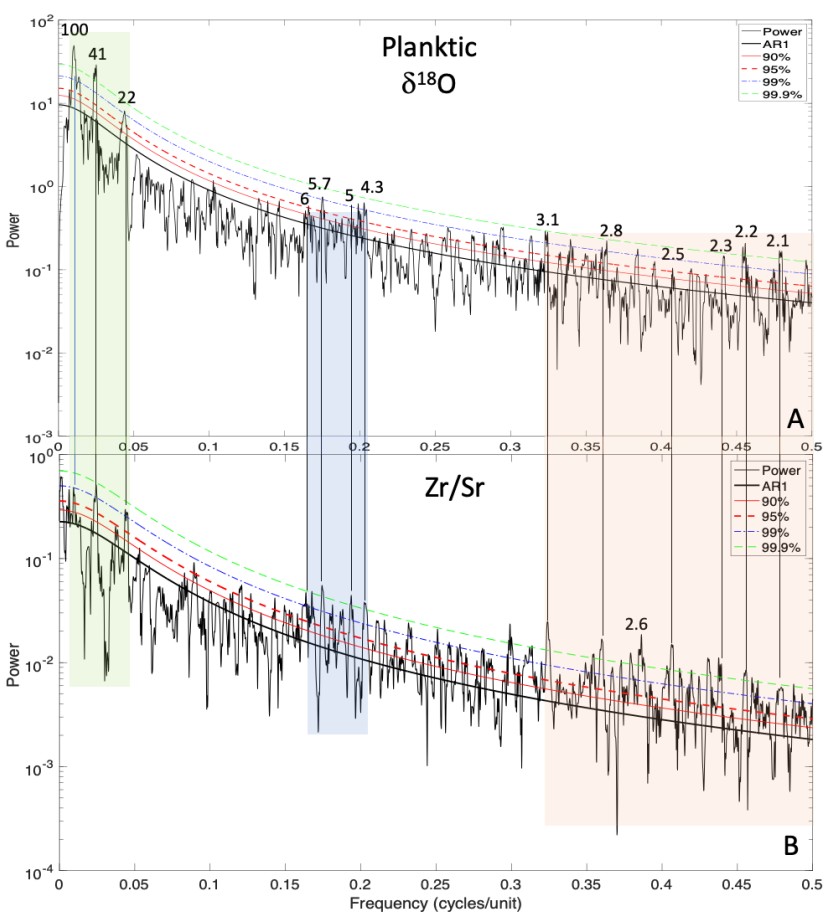

Figure 6. Spectral analysis of planktic δ¹⁸O of *G. bulloides* (A) and Zr/Sr (B) using the multi-
taper method. Signficant peaks shared between the two spectra are labeled with corresponding
periods in kyrs. Orbital bands are highlighted in green. A cluster of peaks occur between 6 and
4 kyrs (blue shade) and the remainder <3 kyrs (red shade).
**3.3 Description of records**
Because it is difficult to distinguish millennial events when the Site U1385 record is plotted
full scale (Fig. 3), we describe the time evolution of orbital and suborbital variability in the
isotope and XRF records for the last 1.45 Ma in ~200-kyr increments: 0-200 ka (Fig. 7); 200-
400 ka (Fig. 8); 400-600 ka (Fig. 9); 600-800 ka (Fig. 10); 800-1000 ka (Fig. 11); 1000-1200
(Fig. 12); 1200-1450 (Fig. 13). We begin with the last 200 kyrs because this is the best known
period for MCV that can be used as a benchmark for comparison with MCV in the older



intervals. Within each interval the record is described oldest to youngest. The records consist
of planktic $\delta^{18}O$, benthic $\delta^{18}O$, benthic $\delta^{13}C$ and Zr/Sr with stadial events identified by the gray
shading. We use a modified version of the isotope nomenclature of Railsback et al. (2015) for
marine isotope stages (MIS) of the last million years and the detrital layer stratigraphy of
Channell et al. (2012) for Heinrich events.
**3.3.1 MIS 1-7a (0-200 ka)**
The interval from 0 to 200 ka consists mainly of the record of MD01-2444 which has been
described in previous publications (Martrat et al., 2007; Margari et al., 2010, 2014; Hodell et
al., 2013). MIS 6 shows a typical pattern of strong MCV at the time of glacial inception
following MIS 7a (Fig. 7). Six millennial events are recognized between ~195 and 155 ka with
a recurrence time ranging from 3 to 7 kyrs (Margari et al., 2010, 2014), which also correspond
with carbon dioxide maxima (Shin et al., 2020). Mininum benthic $\delta^{13}C$ values occur at ~155
ka during event 6vi, which is associated with very cold alkenone SSTs (Margari et al., 2014).
MCV becomes more subdued during the full glacial conditions of MIS 6 following by Heinrich
stadial 11 associated with Termination II. MIS 6 shows a clear pattern of decreasing MCV
during the glacial cycle with suppressed variability at the time of peak glaciation. Loulergue et
al (2008) using ice core methane and Antarctic $\delta^{18}O$ showed a similar pattern of millennial
variability, with 5 interstadial events identified between 190 and 170 ka but only 1 event
between 170 and 140 ka. These patterns are also reflected in marine oxygenation
reconstructions from the Southern Ocean (Gottschalk et al., 2020). The close similarity in
pattern between planktic $\delta^{18}O$ and methane, and between benthic and Antarctic ice $\delta^{18}O$,
continues throughout the record (Wolff et al., 2022).
Low-amplitude MCV occurs during MIS 5e (Tzedakis et al., 2018) and is followed by three
strong stadial events during MIS 5d. MIS 5b is marked by a single prolonged period of stadial
conditions. Millennial events DO 20 and 21, documented in the Greenland ice core, are
recorded on the transition from MIS 5a to 4. MCV was relatively suppressed during MIS4
except for a single event (DO 18). The last glacial cycle is unusual in that it is interrupted by a
long period of strong millennial variability during MIS 3 followed by a decreased amplitude
during the last glacial maximum (MIS 2) between ~27 and 19 ka (Fig. 2). MIS 2 is terminated
beginning with Heinrich stadial 1 which marks the start of deglaciation. Termination I includes
millennial events that occurred during the deglaciation including the Bølling-Allerød
interstadial and Younger Dryas stadial.

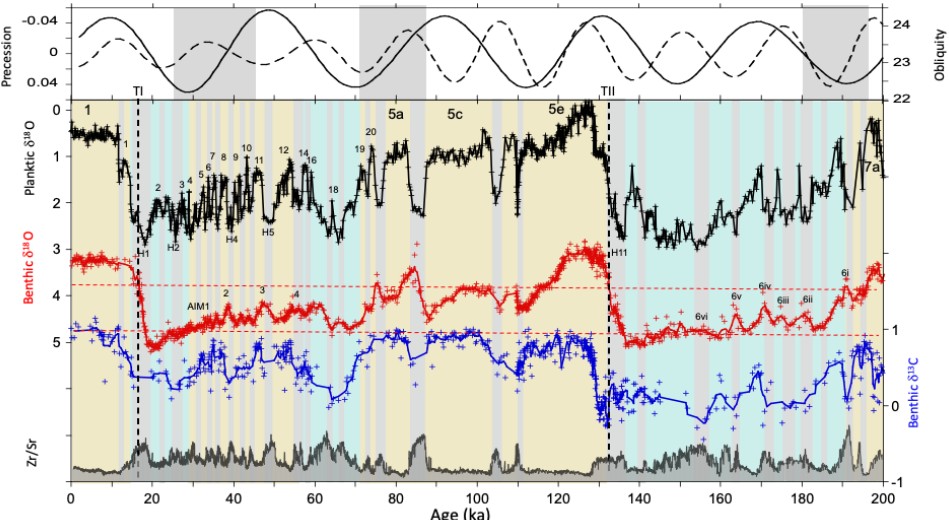

**Figure 7**. 0-200 ka: Planktic δ¹⁸O of *G. bulloides* (black), benthic δ¹⁸O of *Cibicidioides spp*
(red), obliquity (solid line), precession (dashed line), benthic δ¹³C of *C. wuellerstorfi* (blue),
and Zr/Sr (gray). Odd marine isotope stages are numbered and shaded yellow. Glacial periods
are shaded blue with stadial events identified by gray vertical bars. Stadials during MIS 6 are
numbered after Margari et al. (2010). Terminations are indicated by vertical dashed black lines
and roman numerals have been placed approximately near the mid-point of the deglaciation
although millennial events on the termination often make it difficult to exactly define this point.
Horizontal dashed red lines correpsond to the benthic δ¹⁸O thresholds marking the window of
enhanced millennial variability. Precession and obliquity are plotted such that boreal summer
insolation increases in the up direction. The gray shading indicates times when strong MCV is
associated with declining or low obliquity, especially associated with glacial inceptions.

### 3.3.2 MIS 7c-11 (200-400 ka)

The transition from MIS 11 to 10 was marked by strong MCV (Fig. 8) and features in planktic
and benthic δ¹⁸O at Site U1385 can be readily correlated to the EPICA ice core records of
methane and δD, respectively (Nehrbass-Ahles et al., 2020). Initially, the events are paced ~5
kyr apart from 400 to 370 ka and the recurrence time decreases to ~3 kyrs between 365 and
355 ka. MIS 10 culminates in two prolonged Heinrich stadials (10.1 and 10.2) before
Termination IV. Low benthic δ¹³C values (<0 ‰) occur at 338 and 352 ka associated with HS
10.1 and 10.2. MCV is muted during MIS 9a and 9e but relatively strong in the period between
9a and 9e. MCV resumes during MIS 8 including three Heinrich stadial events (8.1, 8.2, 8.3)



prior to Termination III. Minimum δ¹³C values during MIS 8 occur at 272 ka associated with
a very strong cooling event in alkenone SST (Rodrigues et al., 2017). MCV occurs on the
transition from MIS 7e to 7d and is relatively suppressed during MIS 7c.

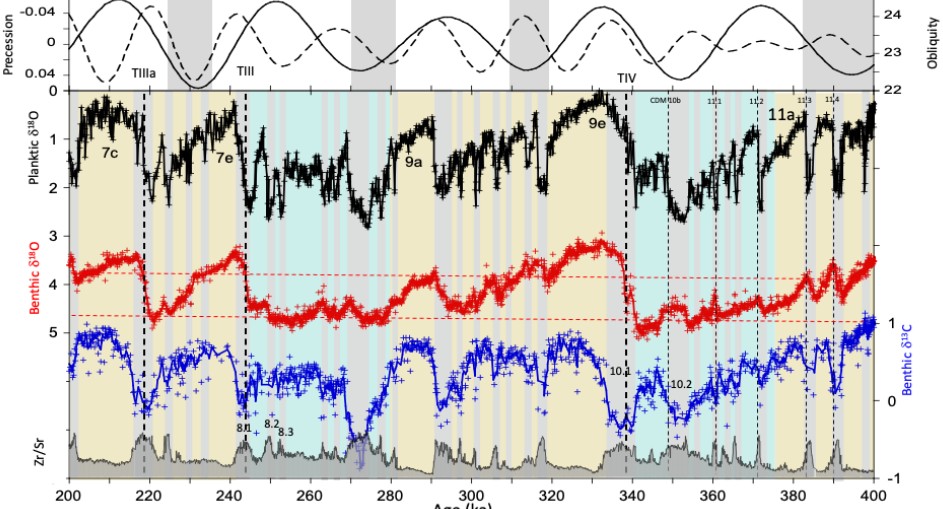

**Figure 8**. Same as Fig. 7 but for 200-400 ka.
**3.3.3 MIS 11c-15c (400-600 ka)**
Both MIS 15d and 15b contain two strong stadial events, whereas MCV was suppressed during
15a, c and e (Fig. 9). MIS 14 was a relatively weak glacial by late Pleistocene standards and
MCV occurred throughout most of the glacial, and especially on the MIS 15a/14 transition.
MIS 13 shows relatively low variability with one stadial event in 13c and two near the 13b/a
transition. Strong MCV is recorded on the glacial inception of MIS 12 followed by a trend of
declining amplitude towards the peak of MIS12. A minimum in benthic δ¹³C values of <0 ‰
occurs in the middle of MIS 12 at 455 ka. A short hiatus (~30 kyr duration) occurs at the
transition from MIS 12 to 11 that removed much of Termination V and early MIS 11.



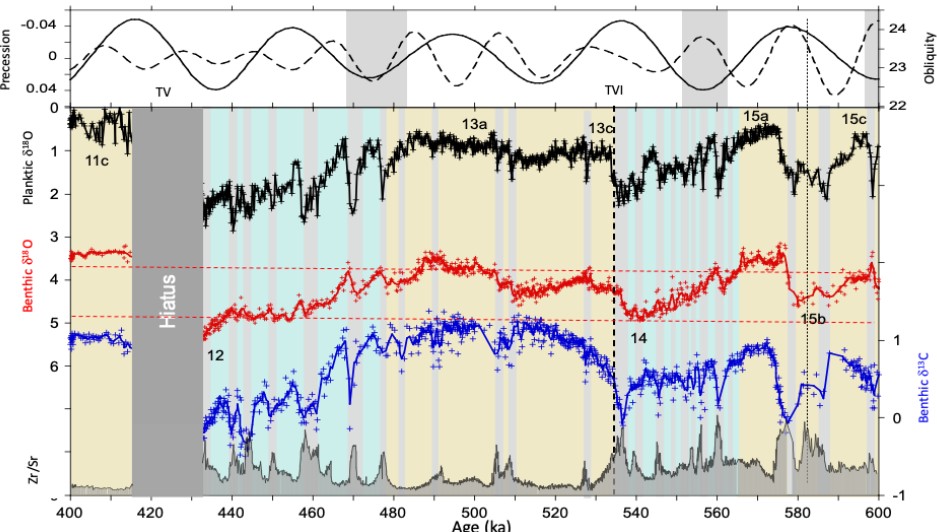

**Figure 9**. Same as Fig. 7 but for 400-600 ka.

**3.3.4 MIS 15e-20 (600-800 ka)**

The end of MIS 20 is marked by a terminal stadial event and decrease in benthic $\delta^{13}$C at 795
ka (Fig. 10). Following MIS 19, strong MCV occurs on the MIS 19/18 transition with three
distinct millennial oscillations paced at ~5 kyrs, which have been interpreted to reflect the
second harmonic of precession (Ferretti et al., 2015; Sanchez-Goni et al., 2016). MIS 19 is the
oldest interglacial recorded in the EPICA Dome C (EDC) ice core and three consecutive
warming events (AIM) occur on the MIS 19/18 transition (Jouzel et al., 2007; Pol et al., 2010),
which were also identified in the CH$_4$ and CO$_2$ signals (Loulergue et al., 2008; Lüthi et al.,
2008). At Site U1385, the phasing between planktic and benthic $\delta^{18}$O variations during the
MCV on the MIS19/18 transition is similar to that observed during MIS 3, suggesting an active
bipolar seesaw (Fig. 2). The phasing between methane and $\delta$D in the EPICA ice core is difficult
to determine because of large uncertainties in gas age-ice age offsets and possible diffusion in
the deepest part of the ice core.

MIS 18 consists of two distinct glaciations separated by a long interstadial period that is
punctuated by a stadial event in the middle at 730 ka. Millennial variability decreases
throughout the glacial inception towards the first glacial peak associated with a decrease in
benthic $\delta^{13}$C at 742 ka. The second glacial peak of MIS 18 is marked by a very strong decrease
in benthic carbon isotope values at 717 ka associated with Terminations VIIIA.



MIS 16 shows a trend of decreasing amplitude of MCV through the glacial cycle where the
variability is greatest on the MIS 17/16 glacial transition and diminishes towards the peak
glacial conditions of MIS 16. Strong stadial events associated with Heinrich events 16.1 and
16.2 are suspiciously absent near Termination VII, perhaps indicating the presence of a
previously undetected hiatus.

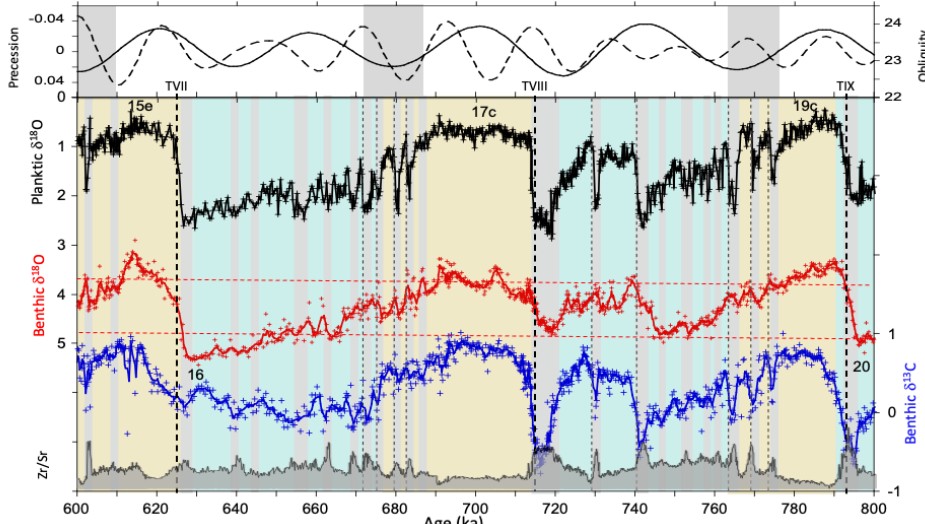

**Figure 10**. Same as Fig. 7 but for 600-800 ka.
**3.3.5 MIS 21-27 (800-1000 ka)**
The pattern of increased MCV associated with the transitions from interglacial to glacial stages
continues with MIS 27/26 (Fig. 11). MIS 26 and 28 were relatively weak glacials and marked
by strong millennial variability. The interval from MIS 25-21 is often compared with MIS 5-1
because of the similarity of weak interglacial MIS 23 to MIS 3. MIS 25-21 is sometimes
erroneously described as the first '100-kyr cycle', but it consists of two obliquity cycles (Bajo
et al., 2020). The MIS 24/23 transition (TXI) was an incomplete (skipped) deglaciation, thereby
lengthening the duration of glacial conditions to ~80 kys. The pace of millennial events is faster
on the MIS25/24 transition than for some other glacial inceptions. Strong MCV is evident
throughout MIS 24 and, unlike MIS 3, MCV is relatively suppressed during MIS 23, which
contains a single strong millennial event at 919 ka. This pattern is different from the last glacial
cycle when MCV was suppressed during MIS 4 and enhanced during a significant portion of
MIS 3.
Glacial ice volume increased (Elderfield et al., 2012) during MIS 25-21and major changes
occurred in deep-ocean circulation (Pena and Goldstein, 2014) and carbon cycling (Thomas



and Hodell, in press). Minimum benthic $\delta^{13}C$ values occurred at 878, 898, and 925 ka, which
are among some of the lowest values found in the deep North Atlantic during the Quaternary
(Raymo et al., 1997; Hodell and Channell, 2016). MIS 21 has multiple substages and consists
of four warm periods that are spaced about 10 kyrs apart, which have been interpreted as the
second harmonic of precession (Ferretti et al., 2010).

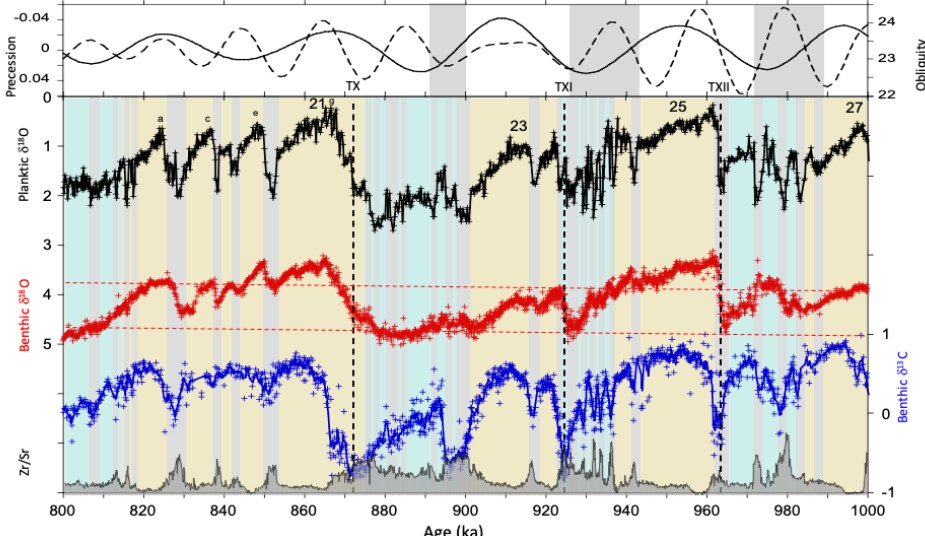


**Figure 11**. Same as Fig. 7 but for 600-1000 ka.

**3.3.6  MIS 28-36 (1000-1200 ka)**

The start of the MPT occurred at 1200 ka (Clark et al., 2006). From 1200 ka onward, the length
of some of the glacial-interglacial cycles increase and the shape of the $\delta^{18}O$ signal becomes
less symmetrical and takes on a decidedly more sawtooth waveform (Broecker and van Donk,
1970), reflecting a slower rate of ice sheet growth and faster rate of decay.

MIS 35-34-33 has a different duration and shape than previous glacial cycles -- it is drawn out
with ~90 kyrs between TXVI and TXIV. MIS 35 was an exceptionally long interglacial
(Shackleton et al., 1990). Very strong MCV occurred on the MIS 35/34 transition (Fig. 12),
consisting of four prominent events that are paced about 5-6 kyrs apart. The abrupt warming
events that mark the start of interstadials coincide with minima in benthic $\delta^{18}O$, indicating that
the phase relationship is similar to that observed during MIS 3 between Greenland and
Antarctica (Fig. 2), which is a pattern indicative of the bipolar seesaw. The benthic $\delta^{13}C$ mirrors
the planktic $\delta^{18}O$ record with strong decreases in benthic $\delta^{13}C$ associated with each of the
stadial events.




The stadial events become progressively colder during MIS 34 culminating in the terminal
stadial event that occurs near the MIS 34/33 transition (TXV). This stadial is marked by a
strong decrease in benthic $\delta^{13}$C. Benthic $\delta^{18}$O begins to decrease first while planktic $\delta^{18}$O
remains high (cold) and benthic $\delta^{13}$C low. This is the same phasing as observed during
Termination I on the Iberian margin when the Southern Hemisphere begins to warm at ~18 ka
as the North Atlantic remains cold and NADW shoals (Skinner and Shackleton, 2006).

Millennial events occur within MIS 33 and on the glacial inception of MIS 32 with a strong
terminal stadial event associated with MIS 32/31 (TXIV). MIS 31 (1094-1062 ka) was also a
relatively long and strong interglacial (Oliveira et al., 2017). MCV occurs on the 33/32, 31/30
and 29/28 glacial inceptions and each is associated with declining obliquity.

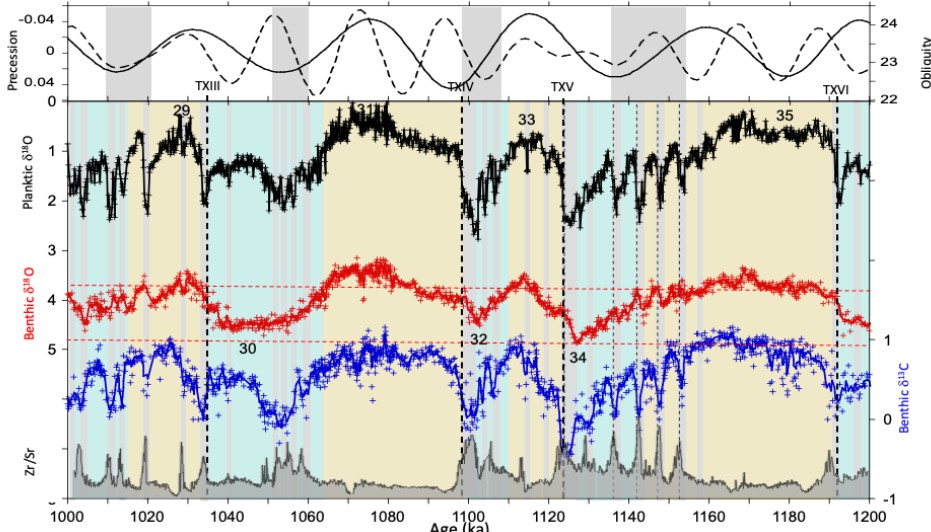

**Figure 12**. Same as Fig. 7 but for 1000-1200 ka.
**3.3.7 MIS 37-47 (1200-1440 ka)**

The period from ~1250 to 1550 ka (MIS 52) in the early Pleistocene was a time when
glacial/interglacial cycles varied dominantly at a 41-kyr period coinciding with variations in
Earth's obliquity, although precession was still significant (Liautaud et al., 2020). Similar to
the last glaciation and Holocene, MCV is enhanced during glacial periods and suppressed
during interglacial stages, exhibiting a threshold response (Fig. 13). MCV increases when
obliquity drops below a threshold value of 23.5°, which corresponds to a benthic $\delta^{18}$O threshold
of ~3.8‰ (corrected to *Uvigerina*). Importantly, and unlike late Pleistocene glaciations after



the MPT, MCV persists throughout most of the glacial part of the cycle. Many of the increases
in planktic $\delta^{18}O$ (stadial events) are associated with coeval decreases in benthic $\delta^{13}C$ indicating
a link between North Atlantic surface climate and deep-water circulation.

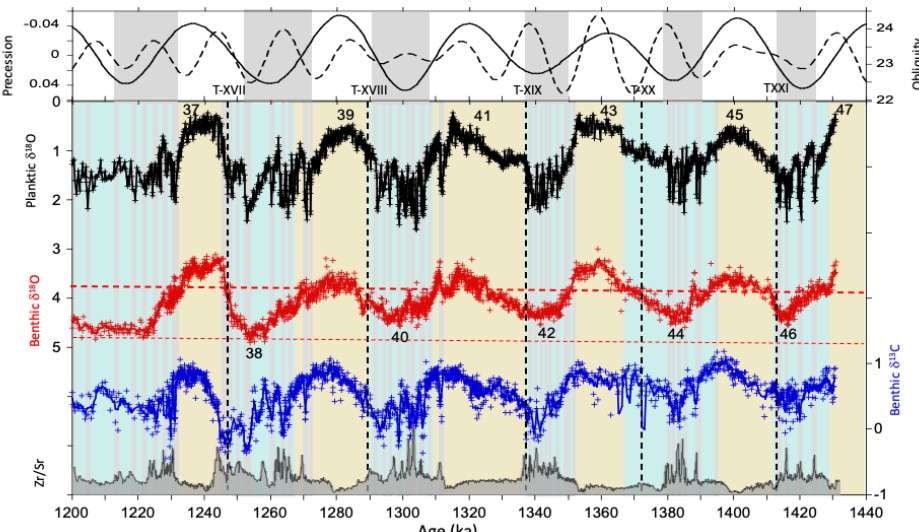

**Figure 13**. Same as Fig. 7 but for 1200-1440 ka.
**4. Discussion**
**4.1 Millennial variability in planktic $\delta^{18}O$**
Because of the great similarity between Greenland $\delta^{18}O$ and Iberian margin planktic $\delta^{18}O$
signals for the last glacial cycle, we interpret this proxy of surface temperature as an indicator
of MCV in the North Atlantic. XRF records of Site U1385 provided the first evidence that
MCV was a persistent feature of the climate system for at least the past 1.45 Ma (Hodell et al.,
2015), which is confirmed by comparison of the planktic $\delta^{18}O$ and Zr/Sr signals (Figs. 4 and
5). The first-order pattern is that MCV is enhanced during glacial stages and diminished during
each of the full interglacial stages (see shading in Fig. 4), which is consistent with the relative
stability of Holocene climate relative to the last glacial period in the Greenland ice core and
with other paleoclimatic results (McManus et al., 1999; Barker et al., 2021; Sun et al., 2021;
Kawamura et al., 2017). A repeated pattern is that the end of each interglacial stage is marked
by the onset of strong MCV that continues through the period of glacial inception when ice
sheets are expanding on North America and Eurasia. MCV associated with glacial inceptions
have generally longer recurrence times than D-O events varying between 3 and 8 kyrs. A few





glacial cycles of the late Pleistocene show a clear pattern of decreasing amplitude of MCV
from the glacial inception towards peak glacial conditions (e.g., MIS 6, 10, 12, and 16, Figs 7-
10), giving rise to a saw-tooth shape. The pattern of MCV evolves from longer stronger
interstadials to shorter weaker interstadials as climate becomes progressively cooler during the
glacial cycle. It is the MCV that consistitute the unevenly-spaced teeth in the saw-tooth pattern
if you will. The last glacial cycle is unusual in that the low MCV during MIS 2 and 4 is
interrupted by a period of strong variability during MIS 3. Such D-O-type MCV has a short
recurrence time (1.5-2 kyrs) which is also found during early Pleistocene glaciations prior to
1.25 Ma (Birner et al., 2016). Almost all glacial periods end with a strong terminal stadial event
that marks the start of deglaciation with some terminations containing additional millennial
events during deglaciation (e.g., Bolling-Allerod and Younger Dryas oscillations).
**4.2 Millennial variability in benthic $\delta^{18}O$**
Unlike planktic $\delta^{18}O$, Shackleton et al. (2000, 2004) demonstrated that variations in the benthic
$\delta^{18}O$ signal of piston cores from the Iberian margin closely follows the $\delta D$ record of Antarctic
ice cores for the last glacial period (Fig. 2). The Site U1385 record indicates that this similarity
extends for the last 800 ka (Fig. 14; Nehrbass-Ahles et al., 2020). The reason for the similarity
of Iberian margin benthic $\delta^{18}O$ and Antarctic temperature is not entirely clear (Skinner et al.,
2007). Shackleton et al. (2000) originally proposed the millennial oscillations in benthic $\delta^{18}O$
during MIS 3 reflected changes in the $\delta^{18}O$ of seawater caused by ice volume variations of the
order of 20 - 30 m of sea level equivalence (Siddall et al., 2008). An alternative explanation is
that millennial variations in benthic $\delta^{18}O$ reflect temperature changes of deep-water. In this
case, the large variations in Antarctic air temperatures are damped by the thermal mass of the
deep ocean and translate into small changes in benthic $\delta^{18}O$, reflecting temperature changes in
the source areas of deep-water formation around Antarctica. The similarity of deep-water
temperature estimated by Mg/Ca at ODP Site 1123 in the South Pacific and Antarctic
temperature (Elderfield et al 2012) supports this interpretation, as does the emerging but sparse
evidence for similarity between mean ocean temperature and Antarctic temperature (Haeberli
et al., 2021). This interpretation implies that surface temperatures in the high-latitude Southern
Ocean were important for regulating deep-ocean heat content, which has implications for deep
ocean circulation and $CO_2$ storage (Jansen, 2018). Skinner et al. (2007) measured benthic
Mg/Ca and $\delta^{18}O$ in core MD01-2444 during MIS 3 and concluded that the benthic $\delta^{18}O$ record
cannot be interpreted as a unique proxy of either deep-water temperature or ice-volume and



must contain a significant local hydrographic component related to the mixing of end member
water masses from the North Atlantic and Southern Ocean which have different $\delta^{18}O$ values.
This is further supported by similar results from the deep Southern Ocean, where benthic $\delta^{18}O$
exhibits a similar (but not identical) pattern to that observed on the Iberian Margin (Gottschalk
et al., 2020), and deep-water temperatures again appear to have decreased during HS4,
consistent with enhanced convection contributing to Antarctic warmth and $CO_2$ rise (Skinner
et al., 2020; Menviel et al., 2015). In all cases, a global glacioestatic signal would only be
transported around the globe on a time-scale that is consistent with ocean transport and mixing
(i.e. centuries to millennia) (Primeau and Deleersnijder, 2009), which would oppose any
proposal of benthic $\delta^{18}O$ tracking global ice volume in synchrony (Gebbie et al., 2012). Indeed,
this is demonstrated by the phasing of benthic $\delta^{18}O$, Antarctic temperature, mean ocean
temperature, and sea level on the last deglaciation (Skinner et al., 2005; Baggenstos et al.,

658    2019).


As in the latest Pleistocene, stadial events are associated with decreases in benthic $\delta^{13}C$ for the
past 1.45 Ma, suggesting that surface coolings in the North Atlantic were associated with
perturbations of deep-water ventilation and carbon storage in the deep-sea (Martrat et al., 2007;
Shackleton et al., 2000; Skinner et al., 2007). Low $\delta^{13}C$ values are associated with each of the
glacial terminations when $\delta^{18}O$ is decreasing and, in some cases, the low $\delta^{13}C$ values are
prolonged and extend into the early part of the interglacial period (Hodell et al., 2009; Galaasen
et al., 2014, 2020).

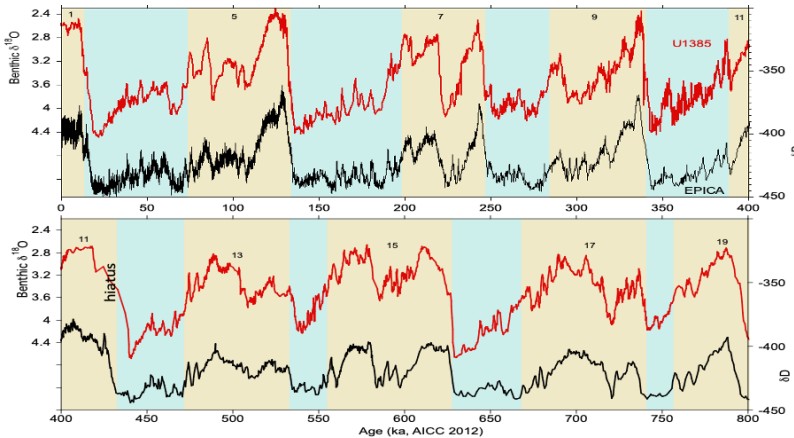

**Figure 14.** Comparison of benthic $\delta^{18}O$ from MD01-2444 and Site U1385 on the Iberian
Margin and $\delta D$ from EPICA Dome C ice core, Antarctica, for the last 800 kyrs.





### 4.3 Phasing of planktic and benthic δ$^{18}$O and the bipolar seesaw

Because of the similarity of the planktic and benthic oxygen isotope records to Greenland and Antarctica, respectively, Shackleton et al. [2000] suggested the relative phasing of inter-hemispheric climate change could be assessed using a single core from the Iberian margin. To evaluate the phasing of MCV, oxygen isotope records of Site U1385 were first high-pass filtered to remove orbital-scale variability. Cross correlation analysis was then performed using the Matlab function xcorr. The filtered planktic and benthic δ$^{18}$O records are weakly correlated and show an average lead of ~0.8 ka for millennial variations in benthic δ$^{18}$O over those of planktic δ$^{18}$O for the past 1500 ka (Fig 15A). The lead of benthic over planktic δ$^{18}$O for the complete record is the same as that observed over shorter time intervals in piston cores during MIS 3 (Shackleton et al., 2000; Skinner et al, 2007) and for Site U1385 during MIS 38 and 40 (Birner et al., 2016). A cross correlation analysis of the isotope records in the depth domain yields a similar result.

This apparent time lead of the benthic over the planktic δ$^{18}$O is more likely the consequence of the different shapes of the benthic (rectangular) and planktic (triangular) signals (Hinnov et al., 2002). Rather than a direct lead/lag relationship between the polar regions, the thermodynamic bipolar seesaw model predicts an anti-correlation between Greenland temperature and the rate of change of Antarctic temperature with the abrupt warmings in Greenland leading the Antarctic cooling onset by about 200 years (WAIS Divide Ice Core members, 2015). This is equivalent to an antiphase relationship between planktic δ$^{18}$O (Greenland temperature) and the time derivative of the benthic δ$^{18}$O signal (Antarctic temperature) from the Iberian margin (Stocker and Johnsen, 2003; Barker et al., 2011). Because taking the derivative of a variable signal can result in noise, the filtered benthic δ$^{18}$O was first smoothed with a 5-point running mean. Although the correlation is poor, the phase shift is as predicted from the thermal bipolar seesaw model (Fig. 15B).

The consistent phase relationships between planktic and benthic δ$^{18}$O during millennial events for the past 1.45 Ma suggest the oceanic bipolar see-saw was a robust feature of the interhemispheric climate system. This phasing is similar despite differences in climate background state; for example, the phasing is the same during glacial inceptions as it is during deglaciations and intermediate ice volume states. Millennial variation in the AMOC and





thermal bipolar seesaw represent mechanisms by which MCV can be propagated to the broader
climate system.

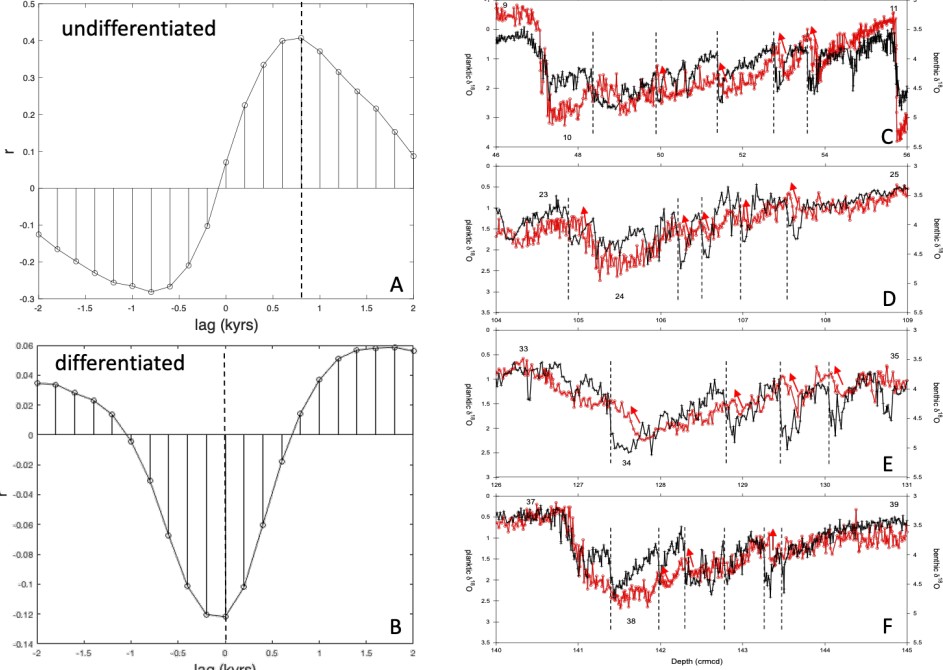

**Figure 15**. (A) Cross correlation coefficient (r) of the filtered signals of planktic and benthic
$\delta^{18}O$. Positive offsets denote a lead of benthic $\delta^{18}O$ over planktic $\delta^{18}O$ by 800 yrs**.** (B) Cross
correlation of planktic $\delta^{18}O$ and the time derivative of smoothed and benthic $\delta^{18}O$. Selected
examples of the phasing of millennial benthic and planktic $\delta^{18}O$ variability in depth domain:
(C) MIS 9 -11; (D) MIS 23-25; (E) MIS 33-35; and (F) MIS 37-39. The vertical dashed lines
mark the rapid warmings (decreases) in the planktic $\delta^{18}O$ record and the red arrows indicate
decreases in benthic $\delta^{18}O$. In most cases, the decrease in benthic $\delta^{18}O$ occurs prior to the
decrease in planktic $\delta^{18}O$, which is similar to the phasing observed during MIS 3 (Figure 2).
**4.4 Orbital modulation of MCV**

To test for amplitude modulation of millennial variability by orbital cycles, we follow the
approach of Hinnov et al. (2002) who analyzed the MD95-2042 record for the last 100 kyrs.
We examine the power spectrum of the planktic $\delta^{18}O$ and Zr/Sr records after applying a Taner
filter and Hilbert transform. Bandpass filtering was performed on evenly resampled (0.2 kyr)
time series using a Taner filter centered on 0.55 ±0.45 with a roll-off rate = $1 \times 10^{12}$, which has





better leakage suppression outside the stopband compared to the Butterworth filter. The
instantaneous amplitude of the modulating signal was calculated by Hilbert transformation.
The presence of significant orbital frequencies in the power spectrum of the Hilbert transform
indicates orbital modulation of the amplitude of MCV, and the evolutive spectra show how the
orbital modulation of MCV has changed through time (Figs. 16 and 17).

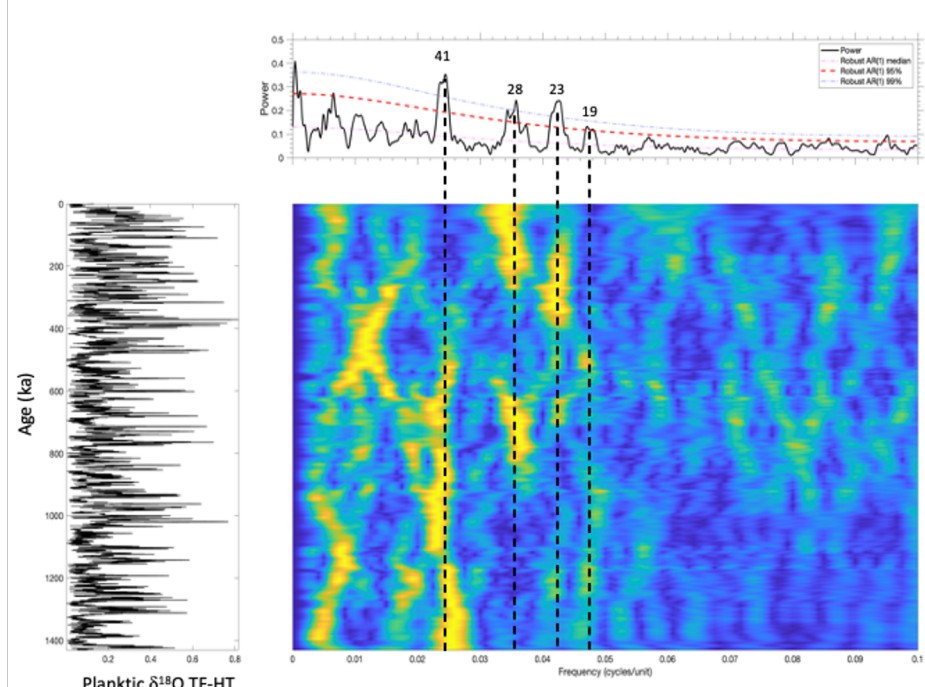


**Figure 16**. Evolutive power spectrum of the amplitude modulation of planktic $\delta^{18}$O as
estimated from a Taner filter (TF) centered at 0.55±0.45 and Hilbert transformation (HT) of
the time series. Sliding window of ~300 kyrs with time domain zero padding and a step equal
to the sampling rate of the time series (~2 kyrs).

The power spectra of planktic $\delta^{18}$O and Zr/Sr are similar and support orbital modulation of the
amplitude of the millennial band by earth's orbital parameters (e.g., 19, 23, 28, 41 kyrs). The
41-kyr obliquity dominates the modulation of MCV between 1450 and ~900 ka with a weak
precession component (Figs. 16 and 17). At ~900 ka, power develops at ~28 kyrs and
precession strengthens, especially in the Zr/Sr record (Fig. 17). The 28-kyr cycle is a common
feature of Pleistocene foraminiferal $\delta^{18}$O records (Huybers and Wunsch, 2004; Lisiecki and
Raymo, 2005; Lourens et al., 2010). The 28-kyr cycle has been interpreted as resulting from



non-linear interactions (combination tones) between eccentricity (quasi-100 kyrs) and
precession (23 kyr) or obliquity (41 kyr). Lourens et al. (2010) suggested the 28-kyr cycle
reflects the sum frequency of the primary 41-kyr cycle and its multiples (82 and 123 ky), and
results from a non-linear response of the glacial cycles to obliquity forcing. At ~650 ka, the
41-kyr and 28-kyr power of obliquity decline substantially and the spectrum is marked by
lower-frequency power (80-120 kyrs). This may reflect an increase in eccentricity modulation
of MCV or modulation by multiples of the obliquity and precession cycles, or a change in the
non-linear energy transfer between orbital components across the MPT (Liebrand and de
Bakker, 2019).

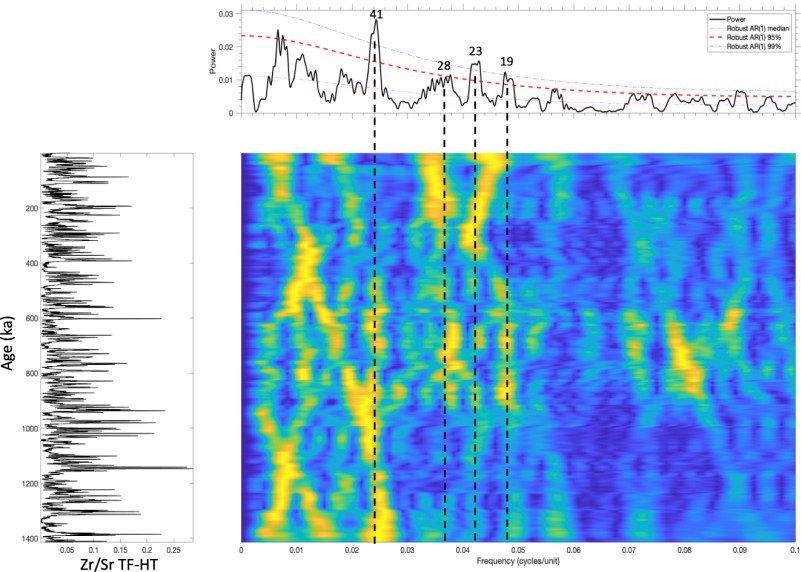

**Figure 17**. Same as Figure 16 but for Zr/Sr.
The early Pleistocene interval from 1200-1440 ka provides strong evidence for a relationship
between the occurrence of MCV and obliquity (Fig. 13). MCV increases during times of low
obliquity, displaying a threshold response such that it increases each time the obliquity drops
below ~23.5°. It is uncertain, however, if the increased millennial variability is the direct (i.e.,
fast-acting) result of reduced insolation at high latitude caused by low obliquity (e.g., lowered
insolation, colder temperature, sea ice expansion) or whether low obliquity secondarily leads
to increased ice volume, ice-sheet height and lowered sea level (see Discussion section 4.5) --

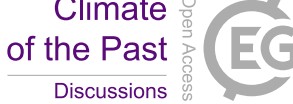



all of which have been proposed as a threshold for MCV (McManus et al., 1999; Zhang et al.,

760     2014).

Some modelling experiments have demonstrated increased MCV during times of low obliquity
in the absence of freshwater forcing (Friedrich et al., 2010; Brown and Galbraith, 2016;
Galbraith and de Lavergne, 2018). The obliquity threshold observed for the early Pleistocene
is highly suggestive of a non-linear system that is influenced by orbital cycles. For example,
sea ice expansion during times of low obliquity may provide strong albedo-feedback
amplification, resulting in a non-linear response (Tuenter et al., 2005). As the mean position of
the sea ice edge expands to lower latitude, the region of deep water formation moves from the
Norwegian-Greenland Sea to south of Iceland, shifting the AMOC with respect to the mean
atmospheric precipitation field where precipitation exceeds evaporation, thereby making the
system less stable (Sevellec and Fedorov, 2015, Friedrich et al., 2010).
The relationship between low obliquity and enhanced MCV persists after 1.2 Ma and is
expressed as increased millennial variability associated with the transitions from glacial to
interglacial stages, which is always associated with declining obliquity (Tzedakis et al., 2012).
In view of this, Tzedakis et al. (2012) proposed the end of interglacials could be defined as
three thousand years (kyr) before the reactivation of MCV at the time of glacial inception. Low
obliquity is important for controlling ice accumulation at the start of a glaciation because ice
growth begins at high latitudes (and altitudes) where the effect of obliquity on summer
insolation is strongest (Vettoretti and Peltier, 2004). Lower obliquity decreases the summer
insolation at high latitudes, reduces seasonality and strengthens the insolation gradient between
low and high latitudes, thereby increasing the meridional heat and moisture flux to the high
latitudes (Mantis, 2011). The increased heat transport does not balance the direct cooling
effects of obliquity through reduced insolation at high latitude. Increased moisture transport
towards the poles provides the fuel needed for growing ice-sheets (Vimeux et al. 1999; Raymo
and Nisancioglu 2003). The combination of reduced temperature and increased moisture flux
are the two ingredients needed for rapid ice sheet growth during glacial inceptions. Precession
and atmospheric $CO_2$ play secondary roles at glacial inceptions that may reinforce or delay
increased ice accumulation depending on $CO_2$ concentration and the phasing of precession and
obliquity (Vettoretti and Peltier, 2004).
Modelling studies suggest that orbital forcing may play a more direct role in the onset of MCV
at the end of interglacial periods. Using LOVECLIM1.3, Yin et al. (2021) found a threshold



response to decreasing summer insolation related to both precession and obliquity. When
summer insolation falls below a critical value, a strong, abrupt weakening of AMOC is
triggered as sea ice expands in the Nordic and Labrador Seas. The transition into a cooler mean
climate state is accompanied by high-amplitude temperature variations lasting for several
thousand years (Yin et al., 2021).
Zhang et al. (2021) used a fully coupled climate model and found that changes in Earth's orbital
geometry can directly affect MCV during intermediate glacial states (e.g., MIS 3). Both
obliquity and precession play a role in AMOC stability (Zhang et al., 2021; Yin et al., 2021) -
- obliquity through its effect on mean insolation at high latitudes and eccentricity-modulated
precession through its low-latitude effect on the subtropical hydrologic budget and salinity of
the North Atlantic basin.
MCV can also result from orbital forcing that is expressed as subharmonics and combination
tones of the primary orbital cycles. Using bispectral analysis, Hagelberg et al. (1994)
demonstrated that approximately a third of the variability in the frequency band ranging from
1/15 to 1/2 kyr originates from the transfer of spectral energy from the lower-frequency
Milankovitch band (see also Liebrand and de Bakker, 2019). A case in point is the 11- and 5.5-
kyr cycles found in MIS 21 and 19, respectively, that have been attributed to the second and
fourth harmonics of the primary precession cycles (Ferretti et al., 2015; Sanchez-Goni et al.,
2016). Berger et al. (2006) suggested the double maximum that occurs in daily irradiation at
tropical latitudes includes a suborbital insolation forcing at 11-kyr and 5.5-kyr periods related
to precession harmonics. Zhang et al. (2021) proposed a physical mechanism for MCV related
to the effect of eccentricity-modulated precession through its low-latitude effect on the
subtropical hydrologic budget and salinity of the North Atlantic basin.
**4.5 State dependence of MCV**
Orbital changes may influence MCV directly through fast processes (e.g., sea ice) or more
indirectly through slow changes in ice sheet configuration (volume or height) and sea level. On
the basis of a 500-ka-long record of ice-rafted detritus and summer SST from Site 980 at 55 °N
in the Rockall Trough, McManus et al. (1999) suggested that MCV was enhanced during times
of intermediate ice volume as defined by a window or "sweet spot" when MCV was most active
during times of intermediate glacial states (Sima et al., 2004; Galbraith and de Lavergne, 2018).
MCV is suppressed under full interglacial conditions and during some peak glaciations. The



concept of increased MCV during times of intermediate ice volume is supported by
observations from the last glacial cycle when MCV was relatively suppressed during MIS 2
and 4 and strong during MIS 3. MCV was also frequent during glacial periods of the early
Pleistocene between 1.45 and 1.25 Ma when glacial benthic $\delta^{18}O$ values fell entirely within the
millennial window (Fig. 5). After 1.25 Ma, the benthic $\delta^{18}O$ threshold is crossed slowly during
glacial inceptions and more quickly at glacial terminations (Sima et al., 2004) with some, but
not all, full glacial periods marked by reduced MCV.

We tested whether there is a statistically significant tendency for millennial events to occur
within a certain range of benthic $\delta^{18}O$ values at Site U1385. The FindPeak algorithm in MatLab
returns the age of each event identified, which is then used to lookup its corresponding benthic
$\delta^{18}O$ value. The $\delta^{18}O$ values are concatenated to form a subpopulation of benthic $\delta^{18}O$ values
corresponding to millennial events that is compared with the full population of benthic $\delta^{18}O$
values (Fig. 18A&B). A two-sample Kolmogorov-Smirnov (K-S) test is used to evaluate if the
two populations are from the same or different continuous distributions and whether the tail of
the millennial subpopulation distribution is smaller than the full population of benthic $\delta^{18}O$
values.

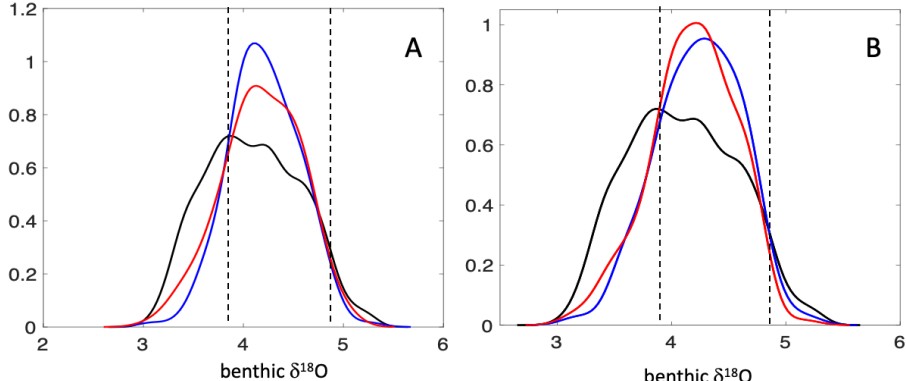


**Figure 18**. Probability density estimate of benthic $\delta^{18}O$ for all values (black), interstadial (red);
and stadial (blue) events for planktic $\delta^{18}O$ (A) and Zr/Sr (B). Vertical dashed lines represent
benthic $\delta^{18}O$ threshold values for MCV that define the millennial window.

For millennial events identified in both planktic $\delta^{18}O$ and Zr/Sr, the millennial benthic $\delta^{18}O$
population was significantly different from the full population at 95% confidence, and the tail
of the millennial populations was significantly smaller than that of the full $\delta^{18}O$ population.



Millennial events are clearly less frequent at the low (warm) end of the benthic $\delta^{18}$O
distribution suggesting reduced MCV during full interglacial periods. We estimate the lower
benthic $\delta^{18}$O threshold to be ~3.8 ‰ for both planktic $\delta^{18}$O and Zr/Sr (note that 0.64 ‰ must
be subtracted from this value to convert to the *Cibicidoides* scale) (Fig. 11C&D). The $\delta^{18}$O
threshold for MCV may differ depending on the record and proxy used to identify millennial
variability (IRD, SST, planktic $\delta^{18}$O, etc.) and may be non-stationary through time. For
example, Bailey et al. (2010) suggested that the $\delta^{18}$O threshold for the late Pliocene (MIS G4
at ~2640 ka and MIS 100 at ~2520 ka) was 0.45‰ lower than that of the late Pleistocene.  For
the period 1240 to 1320 ka at Site U1385, Birner et al. (2016) suggested the threshold was
3.2‰ on the *Cibicidoides* scale which is equivalent to 3.84‰ on the *Uvigerina* scale. This is
the same value we have estimated for the entire 1.5 million yr interval, suggesting the benthic
$\delta^{18}$O threshold has not changed significantly at Site U1385. The existence of an upper $\delta^{18}$O
threshold above which millennial variability is supressed during peak glacial conditions is less
clear from the probability density estimates (Fig. 18). However, several of the late Pleistocene
glacial intervals (MIS 2, 4, 6, 10, 12, 16) show a pattern of strong MCV associated with glacial
inception that decreases towards full glacial conditions (>4.8‰ on the *Uvigerina* scale),
suggesting reduced MCV during maximal glacial conditions.
The physical significance of the benthic $\delta^{18}$O thresholds that define the millennial window is
uncertain. Although several studies have suggested that MCV is related to ice volume, it's not
certain which part of the climate-cryosphere system was responsible. Several processes have
been suggested to trigger increased MCV including sea level dropping below a critical sill
depth (e.g., Bering Sea; De Boer and Nof, 2004), the effect of ice sheet height on winds (Zhang
et al., 2014), iceberg calving and freshwater fluxes to the oceans, or direct orbital forcing
(Friedrich et al., 2010; Zhang et al., 2021; Yin et al., 2021).
McManus et al. (1999) suggested that MCV was enhanced with a sea level lowering of as little
as 30 m below modern. This may correspond to a critical sill depth such as the Bering Sea,
which has a sill depth of ~45 m. De Boer and Nof (2004) proposed that the onset and cessation
of flow through the Bering Strait was responsible for the switch between stable and unstable
states of glacial versus interglacial climate. A restricted Bering Strait increases the sensitivity
of AMOC to freshwater perturbation by blocking the escape route of freshwater to the Pacific
via the Arctic (Poppelmeier et al., 2020; Hu et al., 2012a,b). Freshwater can accumulate in the



North Atlantic more readily with a closed Bering Strait, thereby increasing surface
stratification and leading to AMOC instability.

Because benthic $\delta^{18}O$ also depends on bottom temperature, the threshold could also be related
to surface temperature conditions in the source area of deep-water formation. For example, the
benthic $\delta^{18}O$ threshold could correspond to crossing the freezing point of seawater in deep-
water source areas in the North Atlantic, which would result in increased sea ice formation and
shift the region of deep-water formation to the south where the AMOC is more susceptible to
oscillation (Sevellec and Fedorov, 2015).

Galbraith and de Lavergne (2018) suggested that D-O-like variability in AMOC was more
likely to occur under a 'sweet spot' of interrelated conditions that included low obliquity, low
$CO_2$ and a low-elevation Laurentide ice sheet. By analyzing dust flux from the Dome Fuji ice
core (Antarctica) over the last 720 kys, Kawamura et al. (2017) also concluded that MCV was
more likely during times of intermediate glacial states. Because glacial climate state is
ultimately affected by orbital geometry, an inherent link must exist between climate variability
on orbital and suborbital time scales (see discussion in section 4.4).

**4.6 MCV across the Middle Pleistocene Transition**

The MPT occurred between ~1200 and 650 ka and involved an increase in global ice volume
as the dominant period of glacial-interglacial cycles shifted from 41 kyrs before 1200 ka to
quasi-100 kyrs after 650 ka (Clark et al., 2006). Some studies have suggested that MCV was
less frequent during the early Pleistocene and increased across the MPT as the size of Northern
Hemisphere ice sheets expanded (Larrasoana et al., 2003; Weirauch et al. , 2008; Bolton et al.,
2010). Others have found evidence for equally strong millennial variability in the early
Pleistocene as the late Pleistocene (Raymo et al., 1998; McIntyre et al., 2001; Tzedakis et al.,
2015; Grützner and Higgins, 2010; Hodell et al., 2008; Birner et al., 2016). Still others have
suggested MCV (as represented by IRD events) was more frequent but less intense prior to 650
ka because the climate system spent more time in the millennial window during the early
Pleistocene (Hodell et al., 2008; Hodell and Channell, 2016) and rarely exceeded the upper
benthic $\delta^{18}O$ threshold before 650 ka.

The planktic $\delta^{18}O$ and Zr/Sr records of Site U1385 clearly demonstrate that MCV was strong
during glacial stages both before and after the MPT. The main difference across the MPT is





that whereas MCV persists throughout the glacial periods prior to 1200 ka, it is most prevalent
on the transitions both into and out of glacial states (i.e., inceptions and terminations) and
during times of sustained intermediate ice volume, such as MIS 3. Beginning at 650 ka (MIS
16) following the MPT, MCV is suppressed during some of the strongest glacial periods
associated with the growth of oversized continental ice sheets, which Raymo (1997) refers to
as "excess ice".
A change occurred in the orbital modulation of MCV across the MPT as expressed in changes
in the evolutive spectra of the Taner filter-Hilbert transform of the Zr/Sr and planktic $\delta^{18}O$
(Figs. 10 and 11). Prior to ~900 ka, the amplitude modulation of MCV was dominated by 41-
kyr obliquity. Obliquity continues to modulate the amplitude of MCV from 900 to 600 ka, but
with an increase in precession and the addition of a possible combination tone (28 kyrs) of the
41-kyr cycle.
At the end of the MPT (~600 ka), the power of obliquity declines and the spectra become more
complex with greater modulation at lower frequencies (e.g, 100±20 kyrs). Hodell et al. (2008)
noted a similar change in the amplitude modulation of the Si/Sr IRD proxy at Site U1308 in
the central North Atlantic IRD belt when, at ~650 ka, the power of the 41-kyr obliquity cycle
decreased and quasi-100-kyr power increased. Hodell and Channell (2016) also observed that
millennial variability in the Si/Sr IRD proxy was proportional to the power in the precessional
band, suggesting an amplitude modulation of MCV by precession. Precession plays a greater
role in modulating the amplitude of MCV in the late Pleistocene, in agreement with its steady
increase in importance throughout the Quaternary (Liautaud et al., 2020).
At 0.65 Ma, the development of massive ice sheets on North America (Batchelor et al., 2019)
introduced a new type of MCV related to dynamic instability of the Laurentide Ice Sheet in the
region of Hudson Strait, which was expressed by the occurrence of Heinrich layers in North
Atlantic sediment beginning in MIS 16 (Hodell et al., 2008; Hodell and Channell, 2016).
Heinrich events tend to occur late in the glacial cycle and are associated with glaciations of
long duration (Hodell et al., 2008). They are distinct from background IRD events in their
magnitude, frequency and duration, and their impact on the global climate system was more
widespread (Marshall and Koutnik, 2006). MCV associated with late Pleistocene terminations
after 0.65 Ma are closely related to freshwater fluxes from the decay of oversized ice sheets,
which play an important role in the progression of glacial terminations (Wolff et al., 2009;
Barker and Lohman, 2021).





### 4.7 Influence of MCV on glacial-interglacial cycles

Ice dynamics may be an effective mechanism for propagating high-frequency variability to longer, orbital timescales (Verbitsky et al., 2019). For example, Siddall et al. (2006) suggested that sustained ice-sheet growth through a glacial cycle requires the absence of MCV. Niu et al. (2019) proposed that the presence of strong MCV may prevent an ice sheet from reaching its maximum size owing to surface mass balance effects. If true, then sustained MCV throughout the glacial periods of the early Pleistocene may have prevented ice sheets from growing as large as their late Pleistocene counterparts. In contrast, strong MCV on glacial inceptions may have significantly slowed ice sheet development giving rise to the sawtooth shape of the late Pleistocene benthic $\delta^{18}O$ signal. Ice sheets could only reach their maximum size during the latter part of the glacial cycle once MCV was suppressed.

The exact cause-effect relationship between MCV and ice sheet size is difficult to ascertain. Did ice sheets grow larger in the late Pleistocene because MCV was suppressed or did large ice sheets lead to a suppression of MCV during full glacial conditions? In either case, orbital and millennial-scale variability cannot be considered separately from one another because they interact.Verbitsky et al. (2019) demonstrated that ice sheet non-linearity allows MCV to propagate upscale and influence ice-age dynamics. In addition, non-linear ice-flow dynamics can propagate downscale and affect the millennial part of the spectrum.

High-frequency MCV constitutes a source of noise on orbital time scales, which may enhance the phase-locking of the response of the climate system to orbital forcing (Hodell and Channell, 2016). The theory of stochastic resonance has long been considered as a possible mechanism to explain how the climate system can be synchronized with relatively weak orbital forcing (Benzi et al., 1982). The noise for stochastic resonance is often assumed to be random and white, but MCV provides a source of noise to the climate system whose amplitude varies with climate background state -- i.e., relatively noisy glacials and quiet interglacials. Such oscillations in noise amplitude may be relevant for stochastic or coherence resonance in which the signal–noise resonance is important for phase locking (Liu and Chao, 1998). For example, glacial-interglacial variations during the early Pleistocene may consist of a resonant system in which the intensity of millennial variability is responding to obliquity-controlled changes in climate background state and, in turn, changes in the amplitude of MCV may aid in phase locking the climate system to the obliquity period.



### 4.8 MCV and atmospheric $CO_2$ variations

Because nearly all the rapidly exchangeable carbon in the ocean-atmosphere system is
contained in the deep ocean, atmospheric greenhouse gas variations in ice cores are intimately
linked to carbon storage in the deep ocean. Variations in benthic carbon isotopes at Site U1385
demonstrate that the millennial changes in planktic $\delta^{18}O$ are not only a feature of surface
climate on the Iberian margin, but are crucially linked with changes in deep-water ventilation.
Decreases in benthic $\delta^{13}C$ are associated with increases in planktic $\delta^{18}O$, indicating reduced
ventilation of the deep North Atlantic during cold stadial events. A relationship between
atmospheric $CO_2$ and centennial-millennial events in the North Atlantic exists for the last
glaciation and deglaciation (Marcott et al., 2014; Bauska et al., 2021) as well as for older
periods such as MIS 6 (Shin et al., 2020) and the MIS 11-10 transition (Nehrbass-Ahles et al.,

1007 2020).


We suggest MCV may be involved in setting the minimum $CO_2$ values attained during glacial
periods. Millennial variability in AMOC provides a mechanism by which deep-sea $CO_2$ can be
degassed to the atmosphere. When MCV was strong during MIS 3, $CO_2$ varied between 200
and 220 ppm and the lowest sustained $CO_2$ levels of 180-190 ppm were only achieved during
MIS 2 when MCV was suppressed during peak glacial conditions. By analogy with MIS 3, the
persistently strong MCV that occurred throughout the glaciations of the early Pleistocene
(1.45-1.25 Ma) may have prevented $CO_2$ from reaching values as low as those attained during
the late Pleistocene because $CO_2$ was episodically released from the deep-sea reservoir during
strong millennial-scale AMOC events. In the early Pleistocene, boron isotope reconstructions
suggest that fluctuations in $CO_2$ varied in phase with obliquity and benthic $\delta^{18}O$ (Chalk et al.,
2017; Dyez et al. 2018). The threshold-type behaviour of MCV during the 41-kyr cycles of the
early Pleistocene may have served as an important mechanism for linking internal climate
dynamics with external astronomical forcing by regulating carbon storage in the deep-sea.

Evidence from Site U1385 for an active oceanic thermal bipolar see-saw during most of the
prominent stadials during glacials of the 41-kyr world (Birner et al. 2016) supports a similar
mechanism of $CO_2$ degassing via the Southern Ocean as that in MIS 3. Although $CO_2$ records
are fragmentary before 800 ka, there is evidence for elevated glacial $CO_2$ with minimum values
of 220 ppm during glacial periods between 1 and 1.25 Ma during the early MPT (Yan et al.,



2019; Higgins et al. 2015; Chalk et al., 2017; Hönisch et al, 2012), and glacial $CO_2$ values may
have been higher still before 1.25 Ma (Yan et al., 2019; Martınez-Botı́ et al., 2015).

We have emphasized the role that MCV may play in setting atmospheric $CO_2$ concentrations
but others have suggested that, in contrast, atmospheric $CO_2$ may have a controlling influence
on millennial-scale climate oscillations (Zhang et al., 2017; Vettoretti et al., 2022). Using an
Earth system model, Vettoretti et al. (2022) demonstrated that nonlinear self-sustained climate
oscillations appear spontaneously within an intermediate window of glacial-level atmospheric
$CO_2$ concentrations between ~190 and 225 ppm.
**Conclusion**

The recognition of MCV in Greenland ice cores in the early 1980s ushered in the study of
paleoceanographic records at a resolution that is at least 10 times greater than previous orbital-
scale studies. Although the initial focus was on the last deglaciation and MIS 3, several long
records of MCV are beginning to emerge (Hodell et al., 2008; Hodell and Channell, 2016;
Hodell et al., 2015; Barker et al., 2021, 2022), thereby providing an opportunity to document
the long-term relationship of climate variability on orbital and millennial timescales and their
interacttions. Consistent with previous findings, the U1385 record demonstrates that MCV was
a persistent feature of intermediate glacial climate states for the last 1.45 Ma, including the 41-
kyr world of the early Pleistocene prior to the MPT.

During glacial periods from 1.45 to 1.25 Ma, the amplitude of MCV was strongly modulated
by changes in Earth's obliquity and exhibited threshold behaviour typical of a non-linear
system. Beginning at 1.2 Ma at the start of the MPT, MCV becomes more focused on glacial
inceptions, terminations and periods of intermediate ice volume. One of the recurrent patterns
is that strong MCV almost always occurs at glacial inception and continues through the period
of ice growth under conditions of declining insolation forced mainly by obliquity and
secondarily by precession and $CO_2$. During the MPT (1.2-0.65 Ma), obliquity continues to
influence MCV but in a non-linear fashion evidenced by the appearance of combination tones
(28 kyrs) of the 41-kyr cycle (Figs. 9 and 10) in the power spectrum of MCV amplitude
modulation. Near the end of the MPT at 650 ka, MCV amplitude modulation by obliquity
wanes as quasi-periodic 100 kyr and precession power increases. Precession plays a greater
role in modulating the amplitude of MCV in the late Pleistocene consistent with the steady
increase in precession power throughout the Quaternary (Liautaud et al., 2020).



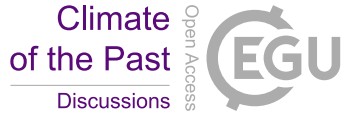


Dansgaard-Oeschger events during MIS 3 are the archetypal example of millennial variability
and considerable effort has been directed towards documenting these events globally, including
the use of numerical models to understand their cause(s). MIS 3 is exceptional relative to the
other latest Pleistocene glaciations in terms of the high number of millennial events and there
appears to be no period like it during the past 1200 ka. The strong, continuous millennial
variability exhibited during MIS 3 is more similar to the millennial variability observed during
glacial cycles of the early Pleistocene from 1440 to 1200 ka (Birner et al., 2016). This similarity
is not entirely unexpected considering that benthic $\delta^{18}O$ values were about the same during
early Pleistocene glacial stages as those during MIS 3, indicating the climate system spent a
prolonged time in an intermediate glacial state. Our analysis of MCV at Site U1385 supports
the concept of a millennial window or sweet spot defined by a lower benthic $\delta^{18}O$ threshold of
~2.9 ‰ below which MCV is suppressed during full interglacial conditions. The upper benthic
$\delta^{18}O$ threshold is less robust despite the fact that some glacial cycles in the late Pleistocene
show a clear pattern of reduced amplitude of MCV as the glacial maximum is approached.
Although the exact physical significance of the benthic $\delta^{18}O$ threshold remains uncertain with
many candidates (ice volume, ice height, sea level, sea ice, deep-water temperatures, etc.),
MCV is strongest during intermediate glacial states.

Climate variability on orbital and suborbital time scales are coupled and interact in both
directions. An example of downscale interaction is the modulation of the amplitude and/or
frequency of MCV by Earth's orbital configuration either through the direct effects of
insolation or more indirectly through ice sheet growth. Some MCV may also be related to
harmonics or combination tones of the orbital cycle (Hagelberg et al., 1994). MCV can exert
an upscale influence on orbital times scales through its effect on ice sheet dynamics (Verbitsky
et al., 2019) or on atmospheric $CO_2$ by changing carbon storage in the deep-sea. MCV is also
a source of noise on glacial-interglacial timescales that may affect the resonance of internal
climate change with external orbital forcing.

In addition to documenting MCV, the planktic and benthic isotope records from Site U1385
provide unprecedented detail of the amplitude and shapes (waveforms) of the glacial cycles on
orbital time scales for the last 1.45 Ma. We emphasize our record is from a single site and
should be compared with other records from higher latitude in the North Atlantic (e.g., Barker
et al., 2021, 2022) and elsewhere (Sun et al., 2021) in order to map geographical differences



over time and develop confidence in the palaeoceanographic interpretations set out here. This
study is also limited to the last 1.45 Ma and we cannot determine the extent to which MCV
was present during glacial periods beyond this time. One of the objectives of upcoming IODP
Expedition 397 is to return to the Iberian margin and extend the U1385 record to study how
orbital and millennial variability co-evolved during through the Quaternary and Pliocene
(Hodell et al., 2022). Understanding these interactions of climate on orbital and suborbital time
scales will lead to a fuller understanding of the mechanisms responsible for the Quaternary ice
ages.

**Data availability**
All datasets and age models have been deposited with PANGAEA as "in review' and will be
publically available when the paper is accepted and a DOI is issued.

**Author contributions**
DAH led the effort to drill Site U1385 and LL and DAH were shipboard scientists aboard IODP
Expedition 393 that recovered the cores. LL constructed the spliced composite section for Site
U1385. MJV provided taxonomic expertise and MJV and NT selected foraminifera and
prepared samples for stable isotope analysis. JER and JN operated the mass spectrometers and
produced the stable isotope data. LL, SJC and DAH oversaw the XRF anlyses of the cores.
LCS, PCT and VM provided data and interpretaiton of Core MD01-2444. EWW advised on
the correlation of the marine sediment record to the Greenland and Antarctic ice cores. DAH,
PCT and EWW wrote the first draft and all authors contributed to the submitted manuscript.
**Competing interests**
Two of the (co-)authors are a member of the editorial board of *Climate of the Past*. The peer-
review process was guided by an independent editor, and the authors also have no other
competing interests to declare.

**Disclaimer**
Publisher's note: Copernicus Publications remains neutral with regard to jurisdictional claims
in published maps and institutional affiliations.



**Acknowledgments**

Samples were provided by the International Ocean Discovery Program (IODP). We thank the IODP Expedition 393 drilling crew, ship's crew, and scientific and technical staff of the drillship *JOIDES Resolution* without whom recovering Site U1385 would not have been possible. We thank Jeannie Booth and Ian Mather for laboratory support. This research was supported by the Natural Environmental Research Council Grants NE/J00653X/1, NE/K005804/1, NE/J017922/1 and NE/R000204/1 and Leverhulme Trust Project RPG2014-417.

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
