# Peer review of "A 1.5-Million-Year Record of Orbital and Millennial Climate Variability in the North Atlantic"

_Climate of the Past, 2022_

## Author Response (AR1)

cp-2022-61
Title: A 1.5-Million-Year Record of Orbital and Millennial Climate Variability in the North Atlantic
Author(s): David Hodell et al.
MS type: Research article

**Response to Reviewer 1**

This manuscript presents a very interesting new record from the Iberian margin, with a remarkable signature of the millenial climatic variability. This new data is of critical relevance to the question of rapid variability and certainly deserves being published.

The paper is well written and easy to read. I only have rather minor comments listed below. Overall, the paper is a bit too long and I am not entirely convinced that all the included information or discussions (sometimes redundant) are truly necessary and I would prefer a slightly reduced paper. For instance, the spectra shown on Figures 16 and 17 are raising more question than providing any understanding of the records.

Figures 16 and 17 (now 15 and 16) are important for assessing the interactions between orbital and millennial bands -- i.e., testing whether orbital variations modulate the nature of millennial climate variability and how it evolves across the MPT. The evidence for amplitude modulation of millennial variability by obliquity prior to 1. 2 Ma is clear in the evolutive power spectra. The spectra admittedly become more complex after 1.2 Ma but we shouldn't shy away from this complexity. The interaction between orbital and millennial variability becomes increasingly nonlinear across the MPT with the appearance of combination tones, such as the 28 kyr power which lacks a straightforward astronomical explanation but is an important observation that will stimulate discussion and future work.

We recognize the paper is long but this is why we chose to submit to *Climate of the Past* rather than a shorter-format journal. In addition to presenting the primary data from Site U1385, the paper also reviews the state of knowledge of MCV and how it interacts with orbital forcing. However, we have tried to follow the recommendation of Reviewer 1 by removing redundancies in the manuscript.

Some minor comments:

1- line 458 typo: correpsond -> correspond
   Corrected in revised manuscript.

2 - line 550 and 909: the MPT… starts at 1200 ka, … ends at 650 ka (Clark et al 2006).

Well, this is highly questionable. It depends of the definition of MPT: if defined in terms of dominant periodicity, it depends of your data and your algorithm. On many other points, the paper provides a detailed and balanced view of different other publications, but not here. At least, this could be discussed a bit more.

We agree that the definition of the MPT is dependent upon the proxy signal and method used to define the shift in dominant frequency from 1/41 to 1/100 kyrs. The MPT is one of the most ambiguous and misunderstood terms in paleoclimatology with descriptions varying from an abrupt versus gradual transition and the timing starting from as early as 1500 ka and

ending as late as 600 ka. In Section 4.6, we have acknowledged this ambiguity in the revised manuscript and explicitly state that we are using a specific definition based on analysis of the LR04 stacked benthic $\delta^{18}O$ record by Clark et al. (2006). It's clear in the Site U1385 benthic $\delta^{18}O$ signal (and other sites) that the period before 1200 ka is dominated by the 41-kyr obliquity cycle. The relevant point of this paper is there is still strong millennial variability in planktic $\delta^{18}O$ during glacials periods between 1200 and 1500 ka when obliquity clearly dominates the glacial-interglacial cycles.

3 - line 616 typo= consistitue -> constitute

Corrected in revised manuscript.

4 - lines 717-748. The appearance of a 28k modulation is unexpected. The given explanations are a bit confusing. It could be much easier to simply note that 1/41 + 1/82 = 1/27 therefore a 27ka oscillation could result from a combination of 41 and 2x41ka. (the 3x41ka = 123ka has no role here). Or am I mistaking something? In any case, this is just numerology… and it is not clear whether this helps towards any understanding of the millennial variability.

The emergence of the 28 kyr power admittedly lacks a straightforward astronomical origin but it is not entirely unexpected because it has been widely reported in late Pleistocene ice core and marine d18O records (Lourens et al, 2010). The explanation for the ~28 kyr beat given by Lourens et al (2010) was based on the sum frequencies between the 41-kyr and its multiples of 82-kyr (i.e. 1/82 + 1/41 = 1/27.3) and 123-kyr (i.e. 1/123 + 1/41 = 1/30.8). While the first combination tone occurs mainly during the early Pleistocene (2.4-2.56 Ma) and the latter during the late Pleistocene (<350 ka), the interval between 1.0 Ma and 650 ka where the ~28-kyr cycle becomes more prominent in the planktonic $\delta^{18}O$ and Zr/Sr records of U1385, seems to consist of only the ~82-kyr beat and not a clear 123-kyr signal. So we have excluded the 3x41kyr option for this part of the discussion. However, the so-called difference frequencies between multiples of the 41-kyr cycle and the main (19-23 ky) precession components could also have increased the power in the ~28 kyr period (e.g., 1/21-1/82=1/28.2), but this option was excluded by Lourens et al (2010) since the early Pleistocene benthic foraminiferal $\delta^{18}O$ records did not reveal a clear precession component. More recently applied bispectral analysis techniques showed that a large part of the precession (spectral) energy could have been transferred to the lower frequencies of obliquity and its multiples in the course of the Quaternary, especially during the MPT (Liebrand and de Bakker, CP, 15, 1959–1983, 2019). In this respect, the presence of a strong precession signal in both the planktonic $\delta^{18}O$ and Zr/Sr records of U1385 could be partially responsible for the occurrence of the ~28-kyr beat, but additional bispectral analyses may be required to further unravel these energy transfers. In Section 4.4, we have revised the discussion of the emergence of the 28 kyr power to highlight the observation that obliquity continues to play a role in the modulation of MCV after the MPT.

5 - lines 947-957. This part on Heinrich events is quite interesting, but entirely disconnected from the rest of the paper. May be, it could be interesting to locate these identified H stadials in some way on Figures 7 to 10? In any case, it would be useful to develop this part and further emphasize the differences between the MCV discussed here (mostly DO type events) and Heinrich events seen in some other records.

Heinrich events (*sensu stricto*) are specific types of millennial events that can be traced to ice surges through Hudson Strait. They occur within certain stadial events (termed Heinrich stadials) but are likely shorter lived than the duration of the stadials in which they are found. The Heinrich stadials are identified on Figures 7 to 10 (now 6 to 9) but this was not clear in the figure caption. The Heinrich stadials have been properly labelled in the figures and explained in the caption.

6 - lines 980-984-987-..1089: "noise". I do understand the discussion of the authors on the role of MCV to trigger transitions and interact with glacial cycles. Still, the word "noise" is certainly fully inappropriate. In mathematics, it is (by definition) a random variable - usually with no temporal structure. In physics, it is (by definition) not a signal "of interest". The millennial variability certainly does not fall in either definition. Please just call it "rapid variability" to avoid misinterpretations.

We admit that "noise" was not the best choice of words to describe the millennial variability. Noise is the term used to describe stochastic resonance or coherence resonance which is why we described it as such. In Section 4.7, we have now callled it "high-frequency variability" and have explained its relevance to "noise" when considering resonance. The idea is that millennial events add "higher frequency variability" to the climate system and the amplitude of that variability changes with glacial and interglacial climate states.  Glacials are more active in the millennial band whereas interglacials are less so. We suggest that such amplitude variations of the high-frequency variability may play a role in stochastic or coherence resonance by phase locking the climate system to the orbital forcing. This hypothesis obviously needs to be tested with climate models and we hope our paper will stimulate further work in this area.

**Response to Reviewer 2**

The manuscript of Hodell et al provides not only a set of high-resolution data for the last 1.5 Ma from a climate sensitive region, but also a quite comprehensive, up-to-date and insightful review for the MCV. The characteristics of the MCV and their links with orbital forcing, glacial state and CO2 are discussed for different periods of the 1.5 Ma. The discussions are made not only based on the U1385 data but also on a wide collection of important literature. I also agree with the authors that "*In addition to documenting MCV, the planktic and benthic isotope records from Site U1385 provide unprecedented detail of the amplitude and shapes (waveforms) of the glacial cycles on orbital time scales for the last 1.45 Ma*". I believe it will be a very useful paper for researchers and students in paleoclimate study. I would recommend its publication after some minor revision. Please find here under my comments and questions which I hope will help to clarify a few things and potentially make the paper more attractive.

1. Figure 6 seems very noisy. There are many peaks between the blue and red shade. I wonder why these are not discussed in the paper. If these peaks are considered as noise, how can we tell the blue and red shaded parts are not noise? I would suggest to try with another software to check whether the result of the spectral analysis is affected by the software used.

   The inaccuracies of the age model render the power spectra difficult to interpret because the peaks in the millennial band are smeared over a wide range of frequencies, resulting in a noisy spectrum. As the reviewer points out, this makes it difficult to distinguish

which peaks are significant and which are not. We have tried different methods of spectral analysis and the results are similar. We have thus removed Figure 6 and the relevant discussion from the paper.

2. The evolutive power spectrum figures (Figures 16 and 17) are not sharp nor nice, which is a pity for such a nice paper where all the other figures are of very high quality. A color bar is also missing. Similar to my comment for Fig.6, I wonder whether the result is affected by the software used, and I would suggest to test with another software to perform wavelet analysis and compare with the results shown in Figures 16 and 17 and to improve the quality of the figures.

The quality of Figures 16 and 17 (now Figures 15 and 16) was severely degraded upon conversion of the manuscript to pdf. The originals of the evolutive power spectra are much clearer and these high-resolution figures will appear in the published paper. We have also added a colour bar as suggested.

3. Lines 733-736: In addition to the 41-kyr signal, Figures 16 and 17 also show strong signal around the low-frequency 0.005-0.01 (200 - 100 kyr cycles) between 1450-900 ka, but I don't see this is mentioned anywhere in the manuscript. Could the authors comment on this and add some discussions in the paper?

We didn't comment on the 200-100 kyr cycles because the periods are long relative to the size of the window (300 kyrs) used for the evolutive spectra. We have noted the longer periods in the text but caution they may not be significant given there are only 1-3 cycles in a given window.

4. The 28-kyr cycle has been suggested to originate from the non-linear interactions between eccentricity and precession/obliquity, or between the 41-kyr cycle and its multiples or from a non-linear response of the glacial cycle to obliquity (lines 32, 739-748, 934, 1056-1058). I would like to draw attention that 28 kyr is one of the important periodicities of obliquity although its magnitude is only one eighth of the magnitude of the 41-kyr periodicity (see Table 1 of Berger, 1978, Journal of the Atmospheric Sciences), so the possibility of a direct and more linear response to obliquity can not be excluded, although the reason why the 41-kyr is switched to 28-kyr at ~800 ka BP remains to be explored. I would suggest to mention this in the manuscript to open more possibilities.

We have mentioned the theoretical obliquity signal contains a secondary peak at ~29 kyr. The spectral power is rather weak, however, so it is unlikely to any direct climatic significance. See also response to comment of Reviewer 1 about the 28-kyr cycle. In Section 4.4, we have rephrased the discussion about the ~28kyr modulation following the suggestions made by both reviewers.

Lines 752-753: I wonder to which extent the good relationship between obliquity and MCV depends on the way to build chronology, and to which extent the obliquity threshold 23.5 is affected by age uncertainty. Could the authors add some discussion on this? The obliquity minimum around 1180 ka seems not playing a role, what might be the reason?

The chronology was built by tuning the colour (L*) to precession which should be independent of obliquity, which was not used in the orbital tuning procedure (Hodell et al.,

2015). We have stated in Section 4.4 of the paper.The obliquity minimum around 1180 ka falls in the middle of MIS 35, which has always been a puzzle. Shackleton et al. (1990) assigned Stage 35 to two obliquity cycles. Although it is possible that the lack of response to the obliquity low at 1180 ka indicates a problem with the time scale, nobody has found a better alternative to Shackleton's proposal that MIS 35 spans two obliquity cycles.

Line 803:  It would be more accurate to say "obliquity through its effect on the mean insolation but mainly on the total summer insolation at high latitudes (see Berger et al., 2010, Quaternary Science Reviews, https://doi.org/10.1016/j.quascirev.2010.05.007), because a large part of the mean insolation is depending on precession.

Yes, that's a very good point. We have changed the sentence as suggested.

7.   Lines 816-818:  It is a repetition of lines 804-805, not related to the subharnomics as discussed in this paragraph, and is better to be integrated with lines 800-805.

Agreed. We have deleted this sentence as it repeats the information in the previous paragraph.

8.   Line 932: Figs. 10 and 11 should be Figs. 16 and 17?

The figure calls have been corrected.

9.   Zr/Sr is less familiar at least for me. Adding some explanation on its paleoclimate interpretation would be welcome.

We added an explanation of why we chose Zr/Sr to represent stadial events in the Methods section (2.1).

10.    Both ka and kyrs are used through the paper. Better to use only one?

We have followed the guidelines provided by the journal for the use of ka and kyr:

"ka stands for "kilo-annum" and literally means thousands of years ago, thus referring to a specific time/date in the past measured from now. In contrast, "kyr" stands for thousand of years and is used to reference to duration."

**Other relevant changes made to the manuscript:**

We replaced Figure 15 (now 14) and eliminated the cross correlation analysis. The correlation coefficients were weak (and likely not significant) and the age model doesn't permit a precise estimate of the phase lag between planktic and benthic d18O signals or its derivative. We have chosen instead to make the point of the similar phasing of older millennial event to MIS 3 by showing several examples from time periods representing different climate boundary conditions (Figure 14).

We have removed the images of *G. bulloides* and *C. wuellerstorfi* from Figure 3. This avoids any copyright issues with the images and is more accurate because benthic d18O was not entirely measured on the species C. wuellerstorfi.

All references have been checked and updated.

The citation and reference for the archived data submitted to Pangaea has been added to the Data Availability statement.